# Multi-omics and network pharmacology identify IGFBP1 as an m⁶A-Epigenetic target of pueraria in NSCLC therapy

Rui Li, Dong-Mei Hu, Yong-Li Liu, Wei Zhao, Yu-Xin Zhang, Yi-Qing Qu 🆔*

Department of Pulmonary and Critical Care Medicine, Qilu Hospital of Shandong University, Jinan, China

* quyiqing@sdu.edu.cn

## Abstract

The dysregulation of N6-methyladenosine (m⁶A) modification drives progression in non-small cell lung cancer (NSCLC), yet its interplay with traditional medicine-derived therapeutics remains largely unexplored. We propose a novel strategy that integrates m⁶A-based prognostic subtypes with Pueraria pharmacology to identify prognostic markers and therapeutic targets related to m⁶A regulators for NSCLC treatment. Multi-omics clustering of 1,661 NSCLC samples identified three distinct m⁶A modification patterns. Based on these, a robust 19-gene prognostic signature was constructed via Cox regression and validated in the GSE31210 dataset. This risk model significantly correlated with immune infiltration and patient survival. Furthermore, the expression patterns of these genes were validated via single-cell RNA-sequencing (scRNA-seq) and RT-qPCR in NSCLC cell lines. To identify pharmacological interventions, we intersected the m⁶A prognostic signature with 7,333 NSCLC-related genes and 366 Pueraria targets, revealing IGFBP1 as the core therapeutic nexus. Immunohistochemistry confirmed the expression of IGFBP1 in NSCLC tissues. Molecular docking and 100-ns molecular dynamics (MD) simulations confirmed stable binding of Pueraria compounds to IGFBP1, specifically 7,8,4'-trihydroxyisoflavone (binding energy = -8.3 kcal/mol) and genistein (-7.4 kcal/mol). This study establishes IGFBP1 as a therapeutic nexus connecting m⁶A-driven NSCLC progression and the anti-tumor effects of Pueraria. Our RNA-modification-guided pharmacology approach advances the integration of traditional medicines into precision oncology.

## Author summary

We investigated how N6-methyladenosine (m⁶A), a prevalent RNA chemical modification, influences non-small cell lung cancer (NSCLC) progression and explored whether natural compounds from the medicinal plant Pueraria could target these pathways. Through the analysis of clinical multi-omics datasets and single-cell RNA sequencing (scRNA-seq), we identified a 19-gene

**Data availability statement:** The data that support the findings of this study are openly available in the TCGA repository (https://portal.gdc.cancer.gov/) (TCGA-LUAD, TCGA-LUSC) and the GEO database (https://www.ncbi.nlm.nih.gov/geo/) under accession numbers GSE50081 (PRJNA215955), GSE68465 (PRJNA282809), GSE119911 (PRJNA490686), and GSE31210 (PRJNA145977). Supporting structural data can be accessed via CB-dock2 (https://cadd.labshare.cn/cb-dock2/php/index.php) and AlphaFold DB (https://alphafold.ebi.ac.uk/entry/AF-Q6PEY6-F1).

**Funding:** This work was supported by the National Natural Science Foundation of China (82173173 to YQQ). The funder had no role in study design, data collection and analysis, decision to publish, or preparation of the manuscript.

signature linking m⁶A regulation to immune activity and treatment outcomes. By integrating these findings with network pharmacology, we established IGFBP1 as a critical therapeutic nexus bridging m⁶A-related tumor biology and Pueraria pharmacology. Experimental validation confirmed IGFBP1 protein expression in tumor tissues, and molecular docking combined with 100-nanosecond molecular dynamics simulations revealed the stable binding of two Pueraria isoflavonoids—7,8,4'-trihydroxyisoflavone and genistein—to IGFBP1. Collectively, our results highlight a viable strategy for repurposing plant-derived compounds to modulate RNA-modification networks in lung cancer. This study paves the way for preclinical testing of IGFBP1-targeting agents and the development of biomarkers to enhance therapeutic efficacy, including immunotherapy, for NSCLC patients.

## Introduction

Lung cancer remains a leading cause of cancer mortality globally [1,2], with non-small cell lung cancer (NSCLC) representing approximately 80–85% of these instances [3]. Despite advancements in molecular and targeted therapies, the five-year survival rate for NSCLC remains below 20% [4]. Improving prognosis and predicting immunotherapeutic outcomes in NSCLC is thus an urgent need.

Recent advances in technology have uncovered RNA modifications, such as m⁶A, found in specific messenger RNAs (mRNAs) [5–7]. m⁶A affects RNA processing, with enzymes like demethylases ("erasers"), "readers," and "writers" (methylases) involved in its mechanism [8–10]. Advances in high-throughput and epitranscriptomic technologies have accelerated the discovery of functionally relevant m⁶A sites and elucidated the complex regulatory networks of RNA modification [11–14].

M⁶A modification is vital for NSCLC progression, with METTL14 and LINC02747 influencing key signaling pathways [15]. The m⁶A modification of MEG3, aided by HNRNPA2B1, may sponge miR-21-5p, upregulating PTEN and inactivating the PI3K/AKT pathway [16]. Cigarette smoking induces m⁶A modification of DAPK2, promoting NSCLC progression via NF-κB activation [17]. These findings show m⁶A modifications influence NSCLC cell growth, metastasis, and key signaling pathways, suggesting m⁶A modulators as prognostic biomarkers and immunotherapy targets [18–20]. Recent study shows m⁶A modification patterns affect prognosis and immunotherapy response in NSCLC [21]. m⁶A patterns in NSCLC relate to immune infiltration and immunotherapy efficacy, but their spatial heterogeneity and interaction with the TME are not well understood [22]. m⁶A regulators may serve as prognostic indicators and targets for immunotherapy in NSCLC and other cancers.

ScRNA-seq enables detailed gene expression analysis at the single-cell level, crucial for understanding cell population diversity [23]. ScRNA-seq reveals the lung adenocarcinoma (LUAD) invasion trajectory cell atlas [24]. Most studies lack multi-omics validation. ScRNA-seq had transformed our understanding of TME complexity in LUAD invasion.

Network pharmacology and molecular docking technology, as important tools in modern drug research, can systematically reveal the multi-component, multi-target, and multi-pathway mechanisms of drug action, making them particularly suitable for the study of complex diseases and traditional Chinese medicine formulas [25–27]. By constructing a drug-target-pathway network and validating target binding affinity through molecular docking, the scientific rigor and accuracy of drug mechanism research have been significantly improved [28].

This study identified m6A-based prognostic subtypes linked to immune infiltration and drug sensitivity in NSCLC and validated using GSE31210 and scRNA-seq. Meanwhile, we also explored the relationship between m6A modification and herbal pharmacology in NSCLC using multi-omics and network pharmacology. Integration of 7333 NSCLC genes and 366 Pueraria targets highlighted 249 hub genes in key pathways, with IGFBP1 as a central therapeutic target, supported by molecular docking and molecular dynamics of Pueraria ingredients. Our study position IGFBP1 as a crucial target for m6A-based prognostic subtypes in NSCLC therapy, providing new theoretical basis and candidate molecules for precision treatment based on epigenetic modification and the modernization of traditional Chinese medicine.

## Methods

### Ethics statement

This study was performed with ethical approval from the Research Ethics Committee of Qilu Hospital of Shandong University (KYLL-2021(KS)-521), and all participants provided written informed consent following the declaration of Helsinki guidelines

### Data sources and preprocessing

We obtained RNA sequencing data, clinical records, and survival statistics from several databases, including GEO (GSE68465, GSE50081), TCGA (LUAD, LUSC), and UCSC-Xena, encompassing a total of 1,661 non-small cell lung cancer (NSCLC) cases (comprising both LUAD and lung squamous cell carcinoma (LUSC)) along with 127 normal tissue samples. In October 2025, we acquired tissue specimens from 10 LUAD patients and 10 LUSC patients from our hospital for the purpose of conducting immunohistochemistry (IHC) experiments. For the RNA sequencing data, the raw count values were transformed into Transcripts Per Million (TPM). The integration of clinical and mutation datasets was performed based on sample identifiers, and batch effects were mitigated utilizing the ComBat function from the "sva" R package. Additionally, the GEO dataset GSE31210, which includes data from 226 NSCLC patients, was incorporated to validate the prognostic model associated with m6A-related genes. Furthermore, the scRNA-seq dataset GSE119911, which consists of 5,406 cells, was included for single-cell analysis following quality control measures. The data analysis was conducted using R version. 4.4.1.

### Unsupervised clustering and functional analysis

We initially identified 23 m6A modulators and employed unsupervised clustering to investigate m6A RNA methylation patterns, assessing stability using ConsensusClusterPlus. To categorize NSCLC samples according to m6A regulator expression profiles, consensus clustering was executed utilizing the "ConsensusClusterPlus" R package (version 4.4.1). The expression matrix, which was generated from m6A-related gene expression data, had samples represented as columns and genes as rows. The analysis was conducted with a maximum of 9 clusters (max K = 9), applying the Partitioning Around Medoids (PAM) clustering algorithm with a Euclidean distance metric. Each clustering iteration was repeated 50 times (reps = 50) with 80% resampling of samples (pItem = 0.8) and full gene features (p Feature = 1). A fixed seed (123456) was established to guarantee reproducibility. The optimal number of clusters was ascertained based on the evaluation of the consensus matrix, resulting in the selection of three clusters (k = 3) for this study. Memberships within the clusters were extracted from the consensus results and designated with alphabetic labels (A, B, C). A differential

expression analysis of m⁶A modulators across the clusters was conducted using the "limma" package, applying thresholds of adjusted $p$-value < 0.001 and |log2 fold change| > 1. Gene Set Variation Analysis (GSVA) was performed to investigate pathway enrichment, utilizing KEGG and GO annotations from MSigDB, with a significance cutoff set at adjusted $p$-value < 0.05.

## Prognostic genes identification and validation

Cox proportional hazards regression (via "survival" package, version 4.4.1) identified prognostic genes among differentially expressed m⁶A regulators, with $p$-value threshold set at <0.001 to enhance specificity. The prognostic risk score was calculated using a multivariate Cox model derived from training data and validated by ROC analysis (using "survival ROC" and "timeROC" packages, version 4.4.1). Validation the m⁶A prognostic model using GSE31210 Gene expression data from the GSE31210 datasets were preprocessed and normalized using the normalizeBetweenArrays function from the "limma" package in R. For datasets without prior log2 transformation, an automatic log2 transformation was applied after correcting for potential negative values. Clinical survival information, including overall survival time and status, was integrated with the normalized gene expression data. A prognostic risk score for each patient was subsequently calculated based on a predefined gene coefficient model. This involved selecting the expression levels of the model genes, standardizing these values, and computing a weighted sum according to the gene coefficients. The gene expression data were scaled across samples, and individual risk scores were derived by applying the model coefficients to the standardized gene expression profiles. These risk scores were then combined with clinical data, and patients were stratified into risk groups based on the median risk score to facilitate survival analysis.

## Tumor microenvironment cell infiltration and gene expression analysis

TME cell infiltration was quantified using single-sample Gene Set Enrichment Analysis (ssGSEA) with the "GSVA" package. Differential expression of m⁶A-related genes among clusters was evaluated via the "limma" package, with a significance cutoff of adjusted $p$-value < 0.001. Expression differences were visualized using box plots from the "ggpubr" package, and statistical significance was assessed with pairwise t-tests adjusted for multiple comparisons.

## Cluster-based survival and genotype analysis

Kaplan-Meier survival analysis was performed using the "survminer" package, and the log-rank test assessed differences among clusters. Gene expression-based subgrouping was performed using the "ConsensusClusterPlus" R package. The input matrix, processed with "limma" to remove duplicate genes and filter low-expression genes, was subjected to consensus clustering (maxK = 9, reps = 50, pItem = 0.8, pFeature = 1) with the k-means algorithm and Euclidean distance. An optimal cluster number of three was identified.

## Differential and correlation analyses

Differential expression of m⁶A genes among clusters was performed with "limma," with Benjamini-Hochberg adjusted $p$-values < 0.001 and |log2 fold change| > 1. Riskscore was calculated based on multiple regression coefficients from Cox models, and the correlation between riskscore, TMB, and gene expression was evaluated using Pearson or Spearman correlation coefficients ("ggpubr" and "reshape2" packages). Survival differences between high and low TMB or risk groups were visualized via Kaplan-Meier curves, with $p$-values from the log-rank test.

## Immune, clinical subgroup and immunotherapy association analyses

To examine relationships between m⁶A-related risk scores, tumor mutation burden (TMB), gene clusters, and clinical traits, we performed comparative analyses using the ggpubr, reshape2 packages. TMB data, risk score groupings, gene

cluster assignments, and clinical data were merged based on shared sample IDs. Extreme TMB values exceeding the 97.5th percentile were capped to reduce the influence of outliers. For TMB analysis, samples were grouped by risk (low vs. high), and box plots were generated using ggplot2 to visualize TMB differences, with statistical significance assessed via pairwise comparisons. For clinical trait analysis, samples were grouped by a selected clinical trait, and box plots were generated to visualize differences in risk scores across the trait categories, again using ggplot2 and pairwise comparisons to assess statistical significance.

To assess the relationship between m6A-related risk scores and response to immunotherapy, we compared Immune Phenoscore (IPS) values (obtained from the LUAD and LUSC of TCIA database) between m6A risk groups using the ggpubr R package. IPS data and m6A risk score groupings were merged based on shared sample IDs. Violin plots, generated using ggviolin, visualized the distribution of each IPS score (presenting the interquartile range) were classified into high-risk and low-risk groups by median risk score, with added boxplots to display median and quartile values. The significance of differences in IPS between m6A risk groups was determined using pairwise comparisons with statistical significance displayed.

## Tumor immune microenvironment

We examined the tumor immune microenvironment of LUAD and LUSC by analyzing gene expression data, immune subtypes, and stemness scores. To investigate the relationships between immune subtypes, m6A gene expression, tumor microenvironment (TME) scores, and stemness scores in LUAD and LUSC, we integrated data from multiple sources. First, gene expression data was filtered for LUAD and LUSC tumor samples and then preprocessed with the limma R package to average duplicate gene entries. Separately, immune subtypes were determined. Differential gene expression analysis was performed between immune subtypes using the Kruskal-Wallis test, and significant genes were identified based on a $p$-value threshold ($p < 0.05$). Expression data was transformed for visualization, and boxplots were generated to compare gene expression levels across immune subtypes. Next, TME scores (Immune-score, Stromal-score, Estimate-score) and stemness scores (RNAss, DNAss) were integrated with gene expression data. Correlations between stemness/TME scores and gene expression levels were assessed using Spearman correlation. Correlations between drug sensitivity and gene expression levels were evaluated using Spearman correlation, and scatter plots visualizing these correlations were generated, with correlation coefficients displayed.

## Cell culture studies and quantitative real-time detection by RT-qPCR

Cell culture studies involved NSCLC cells (A549, H1299, H1975, H1703, H520, H226) and Beas-2B cells, focusing on gene expression related to drug sensitivity and m6A subtypes. We isolated RNA from Beas-2B and LUAD (A549, H1975, H1299) and LUSC (H1703, H520, H226) cell lines using Fastgen, then performed reverse transcription with the EvoM-MLV RT Kit and RT-qPCR (Quantitative Reverse Transcription Polymerase Chain Reaction) with SYBR Green Premix, using GAPDH as a reference gene for m6A differential gene prediction, with primer sequences synthesized from BioSune and expression levels calculated via the 2−ΔΔCT algorithm. all human cell lines have been authenticated using STR (or SNP) profiling within the last three years. All cell lines used are listed using the official cell line name and its Research Resource Identifier (RRID) as available in the ExPASy Cellosaurus database.

## IHC analysis

Paraffin-embedded-fixed LUAD and LUSC tissue sections were hydrated through graded ethanol series (100%, 95%, 80%, 75%) and washed thrice with PBS. Antigen retrieval was performed using microwave-heated EDTA buffer (pH 9.0). Endogenous peroxidase activity was blocked with hydrogen peroxide, followed by serum blocking. Sections were incubated overnight at 4°C with anti-IGFBP1 antibody (Cell Signaling Technology, 31025T) at a dilution of 1:100. After

washing, they were incubated with HRP-labeled secondary antibody and developed using DAB. Sections were counterstained with hematoxylin, differentiated with 1% hydrochloric acid ethanol for 2 seconds, rinsed, dehydrated through ethanol series (75%, 80%, 95%, 100%), cleared, cover-slipped, and examined under a microscope.

## ScRNA-seq analysis

We analyzed 5,406 single cells from 20 NSCLC samples using scRNA-seq in the GSE119911 dataset, applying the "Seurat" package and conducting correlation analysis on intracellular and mitochondrial sequences. Cells with over 5% mitochondrial content, fewer than 50 sequencing counts, and under 100 gene counts were filtered. The data were standardized with Log Normalize, focusing on 1,500 genes with the highest variation for further examination. UMAP visualization was used for TME single cells, revealing differences in $m^6A$ clusters across populations.

## Network docking analysis

We obtained a comprehensive list of NSCLC-related genes by meticulously taking the union of all NSCLC-related genes sourced from several reputable databases, including GeneCards, the Comparative Toxicogenomics Database (CTD), the Therapeutic Target Database (TTD), and the Online Mendelian Inheritance in Man (OMIM) database, ensuring a thorough and inclusive compilation for our research. We extracted the medicinal ingredients of Pueraria from the herb 2.0 database, selected effective ingredients through Swissadme, identified Pueraria components with molecular weight under 500, fewer than 5 hydrogen bond donors, under 10 hydrogen bond acceptors, less than 10 rotatable bonds, Lipid-water partition coefficient below 5, and ensure high gastrointestinal absorption and blood-brain barrier permeability, and predicted the target genes of the effective components of Pueraria using the SwissTargetPrediction database. Next, we took the intersection of the target genes of Pueraria with genes related to NSCLC to obtain the intersecting genes. We conducted GO enrichment analysis and KEGG pathway enrichment analysis on the intersecting genes. Subsequently, we further constructed a regulatory network of Pueraria-effective drug ingredients-target genes-KEGG pathways-NSCLC. We then took the intersection of the target genes of Pueraria with genes related to NSCLC and the prognostic $m^6A$ cluster differential genes obtained from previous screening, ultimately identifying IGFBP1. We then analyzed the molecular docking of IGFBP1 with two effective ingredients of Pueraria using the CB-dock2 website. We used the AlphaFold DB database to download the PDB file of IGFBP1. We used the PubChem database to download the two ingredients of Pueraria.

## MD simulations

We conducted MD simulations on the complexes of 7,8,4'-trihydroxyisoflavone with IGFBP1, as well as genistein with IGFBP1. The simulations were executed utilizing Gromacs version 2022.3. Initially, small molecules were prepared with AmberTools 22, employing the general Amber force field (GAFF). Hydrogen atoms were introduced, and restrained electrostatic potential (RESP) charges were computed using Gaussian 16W. Subsequently, these charges were integrated into the topology files necessary for the MD simulations. The IGFBP1 structure was prepared from the selected PDB, missing atoms/residues were rebuilt, protonation states set for pH 7.0, and the protein topology generated with pdb2gmx using the Amber99sb-ildn force field. Protein–ligand complexes were assembled, placed in a dodecahedral box with ≥1.0 nm buffer, solvated with TIP3P water, and neutralized with Na+ (additional ions added to reach desired ionic strength). Energy minimization used steepest descent until convergence, followed by NVT (100–500 ps, 300 K, V-rescale) and NPT (100–500 ps, 1 bar, Parrinello–Rahman) equilibration with positional restraints, after which 100 ns production runs were carried out at 300 K and 1 bar with 2 fs timesteps, LINCS constraints, PME electrostatics and 1.0 nm cutoffs. Trajectories were saved at regular intervals (coordinates every 10 ps) and analyzed for RMSD, RMSF, hydrogen bonds, and other interaction metrics; all input files and software versions were recorded for reproducibility.

## Statistical analysis

Experiments were done in triplicate; data were analyzed with GraphPad Prism 7.0 and R 4.2.0. One-way ANOVA revealed significant differences among Beas-2B, LUAD, and LUSC cells, with $p < 0.05$ indicating significance.

## Results

### Subtype identification of m$^6$A-related prognostic genes

We constructed an m$^6$A-related prognostic network and identified subtype clusters to explore prognostic implications in NSCLC: the prognostic network of m$^6$A regulators indicates that IGFBP1, HNRNPA2B1, LRPPRC, HNRNPC, YTHDF3, YTHDF1, RBM15B, RBM15, ZC3H13, WTAP, and IGFBP3 serve as risk factors for predicting clinical outcomes in NSCLC. In contrast, FMR1, YTHDF2, YTHDC2, YTHDC1, FTO, METTL3, and IGFBP1 are identified as favorable prognostic factors for NSCLC patients. Furthermore, the network reveals negative associations between the expression levels of FMR1, YTHDC1, IGFBP2, and IGFBP1. Additionally, YTHDF2, YTHDC2, and RBM15 expressions are negatively correlated with the expression of IGFBP3 (Fig 1A). We categorized NSCLC samples into three clusters based on 23 m$^6$A regulators (Fig 1B), revealing that Cluster B had better outcomes than A and C (Fig 1C). Heatmap analysis showed down-regulation of m$^6$A regulators in Cluster A, while B had up-regulation, and IGFBP2 and IGFBP3 were up-regulated in Cluster C (Fig 1D). We categorized our NSCLC patients into three m$^6$A clusters according to gene expression. We conducted differential analyses between m$^6$A clusters B and A, B and C, as well as C and A, using a $p$-value threshold of less than 0.001. The intersection of these analyses revealed a total of 985 that exhibited differential expression across the m$^6$A clusters (Fig 1I). We conducted KEGG and GO analyses to explore the involvement of differentially expressed m$^6$A cluster genes in NSCLC. The results of the GO analysis indicated a significant enrichment of these genes in processes related to glandular and epidermal development. Additionally, KEGG analysis indicated a significant enrichment of m$^6$A cluster differential genes in the phagosome pathway (Fig 1J-1M).

### Mutation/CNV landscape of m$^6$A regulators in NSCLC

We analyzed 23 RNA modification modulators in NSCLC, focusing on CNA, mutation burden, expression, and survival of m$^6$A regulators, revealing gene alterations like ZC3H13, FMR1, and RBM15 (S1A Fig). CNV frequency analysis indicated gains in most m$^6$A modulators (S1B Fig). The graph showed CNV frequency distribution across 23 chromosomes, with some regulators (FMR1, LRPPRC, YTHDC1, HNRNPA2B1, WTAP, IGFBP1, YTHDF3, IGFBP3, VIRMA, FTO, ZC3H13, METTL3, HNRNPC, METTL16, ALKBH5) larger in high copy number samples and others (RBM15, YTHDF1, YTHDF2, IGFBP2, RBM15B, METTL14, YTHDC2, RBMX) in low copy number samples (S1C Fig). The boxplot illustrated the differential expression of 23 m$^6$A modulators in TCGA-LUAD and LUSC patients. METTL3, LRPPRC, RBM15, VIRMA, YTHDF1, HNRNPC, YTHDF2, RBMX, HNRNPA2B1, IGFBP2, IGFBP3 exhibited higher expression in NSCLC tissues, whereas METTL14, METTL16, and ZC3H13 showed significant decreases (S1D Fig). We analyzed 1,037 NSCLC samples and 108 normal tissues from TCGA, plus 181 NSCLC from GSE50081 and 443 NSCLC with 19 normal from GSE68465, resulting in 1,661 NSCLC and 127 normal samples after data cleansing. Survival analysis showed that high expression of FMR1, HNRNPC, IGFBP1, IGFBP3, LRPPRC, RBM15, WTAP, ZC3H13, and HNRNPA2B1 in NSCLC patients linked to poor outcomes, while IGFBP2, METTL3, YTHDC2, and YTHDF2 indicated better outcomes (S2 Fig). High expression of m$^6$A modulators HNRNPA2B1, HNRNPC, IGFBP1, IGFBP3, LRPPRC, RBM15B, and RBM15 in LUAD patients indicates poor survival, while FMR1, YTHDF2, and FTO levels correlate with better outcomes (S3 Fig). Reduced IGFBP1, YTHDF1, and FTO expression in LUSC patients links to better outcomes, while low FMR1, HNRNPA2B1, HNRNPC, IGFBP2, LRPPRC, METTL3, WTAP, and RBM15 levels relate to worse outcomes (S4 Fig). IGFBP1 affects survival in LUAD and NSCLC, but less so in LUSC, indicating m$^6$A modification effects vary by tumor microenvironment and genetics. We used the HPA database to validate m$^6$A regulator expression levels, shown in S5 Fig. In the tissues of

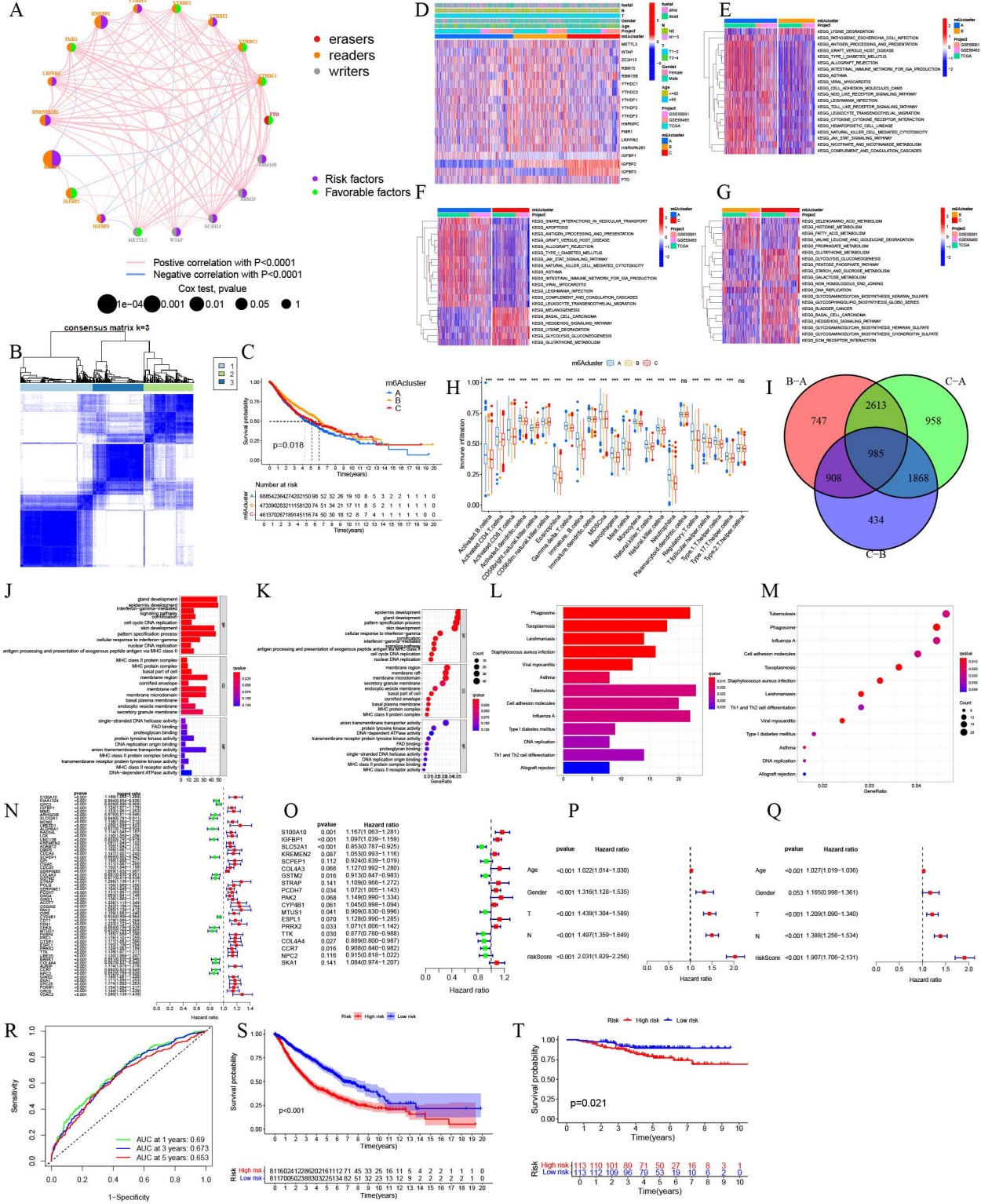

**Fig 1. Multi-omics established m⁶A prognostic clusters and validated clinical relevance A.** In the network, the circles represent m⁶A regulators. The left side indicates different types of m⁶A, while the right side shows associated risk factors, the red lines represent positive correlations, whereas the blue lines indicate negative correlations. **B** The unsupervised clustering analysis was conducted on 23 regulators linked to m⁶A modification in NSCLC.

**C** The examination of survival outcomes concerning the three distinct m⁶A clusters identified in NSCLC. **D** The association between the expression levels of three distinct m⁶A clusters and clinical characteristics in NSCLC. **E-G** The Gene Set Variation Analysis (GSVA) was conducted to compare m⁶A clusters A and B, A and C, as well as B and C. **H** Differential analysis of the infiltration degree of three m⁶A clusters in 23 types of immune cells. **I** Venn diagram of differential genes in m⁶A clusters A, B, and C. **J-M** The boxplot and barplot illustrate the ten biological processes, cellular components, and molecular functions that exhibit the highest levels of enrichment, alongside the thirteen pathways that are most significantly enriched, as determined by the KEGG analysis of differentially expressed genes within the m⁶A clusters. **N-O** Cox regression analyses, both univariate and multivariate, were conducted on the differentially expressed genes associated with m⁶A clusters. **P, Q** Univariate, and multivariate independent prognostic analyzes of m⁶A cluster differential genes. **R** ROC analysis of m⁶A cluster differential genes. **S** Survival analysis of m⁶A cluster differential genes. **T** Survival analysis of the GSE31210 validation dataset.

LUAD and LUSC, high expression levels were observed for the following m⁶A-related regulators: ALKBH5, FMR1, FTO, HNRNPA2B1, HNRNPC, LRPPRC, RBM15, RBMX, YTHDC1, YTHDC2, YTHDF2, and ZC3H13. The complete study is depicted in S16 Fig.

## Pathway enrichment and immune landscape

To clarify the different pathways and functions of the three m⁶A clusters, we used GSVA on datasets from the TCGA, GSE50081, and GSE68465 projects. The heatmap showed that m⁶A cluster C had higher expression levels of lysine degradation and hedgehog signaling compared to clusters A and B. The results of the GSVA analysis indicated that various KEGG signaling pathways are significantly enriched in m⁶A clusters A, B, and C (Fig 1E-1G). The findings regarding the differences in KEGG pathway enrichment differences between m⁶A cluster B, A and A, C, as well as B, C are in the S2-S4 Tables. Additionally, we conducted GSEA analysis using ssGSEA to evaluate immune cell infiltration in 1,622 NSCLC patients across the three m⁶A clusters (A, B, and C). We noted notable variations in the allocation of 21 distinct immune cell types across the m⁶A clusters ($P < 0.05$) (Fig 1H). The immune infiltration patterns of various cell types, including activated B cells, activated CD4 T cells, activated CD8 T cells, activated dendritic cells, CD56 bright natural killer cells, CD56 dim natural killer cells, eosinophils, gamma delta T cells, immature B cells, immature dendritic cells, myeloma-derived suppressor cells (MDSCs), macrophages, mast cells, monocytes, natural killer T cells, natural killer cells, neutrophils, regulatory T cells, T follicular helper cells, type 1 T helper cells, and type 17 T helper cells demonstrated significant variations among the m⁶A clusters A, B, and C. Conversely, no notable differences were observed in the immune infiltration of plasmacytoid dendritic cells and type 2 T helper cells across the three clusters.

## Prognostic model construction and validation

The study identified 57 m⁶A cluster differential genes with a *p*-value < 0.001, subjecting them to multivariate Cox regression analysis (Fig 1N), and eventually selected 19 genes to establish a prediction model (Fig 1O). The predictive risk score was determined by utilizing designated coefficients for each individual gene.The calculation of the survival risk score is performed in the following manner: Survival Risk Score = (0.1543) * S100A10 + (0.0926) * IGFBP1 - (0.1585) * SLC52A1 + (0.0512) * KREMEN2 + (0.0787) * SCPEP1 + (0.1195) * COL4A3 + (0.0915) * GSTM2 + (0.1032) * STRAP + (0.0693) * PCDH7 + (0.1389) * PAK2 + (0.0437) * CYP4B1 + (0.0950) * MTUS1 + (0.1204) * ESPL1 + (0.0690) * PRRX2 - (0.1307) * TTK + (0.1181) * COL4A4 + (0.0965) * CCR7 + (0.0891) * NPC2 + (0.0806) * SKA1. Clinical information and risk data were sourced from GSE50081, GSE68465, TCGA-LUAD, and TCGA-LUSC. Perl and survival R packages were used for independent prognosis analysis. The univariate analysis indicated that age, sex, pathological T and N stages, along with the risk score, act as independent prognostic indicators for patients diagnosed with NSCLC, with a significance threshold of $P < 0.001$ (Fig 1P). Multivariate analysis corroborated age, pathological T and N stages, and risk score as significant high-risk factors ($P < 0.001$) (Fig 1Q). ROC analysis revealed that the AUC values at 1, 3 and 5 years were 0.69, 0.673, and 0.653, respectively, suggesting that the prognostic model based on differential genes from the m⁶A cluster is capable of effectively forecasting the outcomes for NSCLC patients (illustrated in Fig 1R). Furthermore, Fig 1S illustrates

that patients with NSCLC categorized as high risk exhibit a significantly poorer prognosis relative to their low-risk counterparts. In the validation cohorts,we divided the 226 NSCLC patients in the GSE31210 dataset into high- and low-risk groups according to the median risk score. The survival curves revealed that patients in the high-risk group had a poorer prognosis compared to those in the low-risk group, with a statistically significant difference ($P=0.021$) (Fig 1T). The result validate the prognostic utility of the model in independent NSCLC cohorts.

## Genotyping and subtyping based on prognostic genes

We performed genotyping analysis using prognostic $m^6A$ cluster differential gene expression with the "ConsensusClusterPlus" package. This analysis identified three gene clusters: A (n = 179), B (n = 273), and C (n = 179) (Fig 2A). In the genotyping survival analysis, gene cluster C showed a significantly better clinical outcome than clusters B and A ($P=0.007$) (Fig 2B). The genotyping heatmap illustrated that gene cluster A exhibited elevated expression levels of the prognostic $m^6A$ cluster differential genes, in contrast to gene cluster C, which displayed reduced expression levels (Fig 2C). Furthermore, the boxplot analysis revealed notable variations in the expression levels of METTL3, WTAP, YTHDC1, YTHDC2, and YTHDF2 among the different gene clusters designated as A, B, and C ($P<0.001$). We calculated the mean expression of each $m^6A$-related gene in the different gene cluster groups. We observed significant expression differences for METTL3, WTAP, YTHDC1, YTHDC2, YTHDF2, IGFBP2, and IGFBP3 among the three groups, consistent with the boxplot findings (Fig 2D).

## Tumor Mutational Burden (TMB) and immune profiles

Patients exhibiting low TMB demonstrated considerably poorer clinical outcomes in comparison to their high TMB counterparts, with a statistically significant result ($P=0.039$) (Fig 2F). Additionally, the survival analysis revealed that individuals classified with high TMB and low risk scores experienced more favorable clinical outcomes than those with low TMB and low risk scores, as well as those with high TMB coupled with high risk scores (Fig 2E). A high risk score in the prognostic $m^6A$ cluster differential genes was correlated with elevated TMB in NSCLC patients ($P=4.4e-8$) (Fig 2G). Our results indicated that high TMB patients in NSCLC had better clinical outcomes than those with low TMB, suggesting that TMB consistently provides a therapeutic benefit from immune checkpoint blockade (ICB) therapy. The TMB data complements our previous results by highlighting the genetic variability within the tumors, which may influence the efficacy of immunotherapy. Surviving NSCLC patients have lower risk scores than those who have died ($P<2.22e-16$) (Fig 2H). Patients with pathological N1-3 stages in NSCLC have a higher risk score than those with N0 stage ($P=0.0023$) (Fig 2I). NSCLC patients with T1-2 stages display lower risk score than T3-4 counterparts ($P=0.00018$) (Fig 2J). The Immune cell Proportion Score (IPS) in patients with NSCLC exhibited marked disparities between low- and high-risk groups concerning the absence of CTLA4 and PD1 ($P=1.6e-09$) (Fig 2K). Within the low-risk cohort, NSCLC patients demonstrating CTLA4 negativity coupled with PD1 positivity experienced a more favorable response to immunotherapy compared to their counterparts in the high-risk group ($P=6.1e-06$) (Fig 2L). Likewise, low-risk NSCLC patients with CTLA4 positivity and PD1 negativity showed a more favorable immunotherapy response than those in the high-risk cohort ($P=6.6e-11$) (Fig 2M). Furthermore, patients with low-risk NSCLC exhibiting positivity for both CTLA4 and PD1 demonstrated a more favorable response to immunotherapy compared to their counterparts in the high-risk group ($P=9.6e-07$) (Fig 2N). Analysis of risk score and immune infiltration reveals a significant positive correlation with immune cells (Fig 2O).

## Correlation analysis of anti-cancer drug sensitivity and $m^6A$ prognostic genes

An association analysis indicated a positive correlation between the expression levels of CCR7 and the sensitivity to various anti-cancer agents, such as Nelarabine, Fluphenazine, Dexmedetomidine, PX-136, and Chelerythrine ($P<0.05$). Similarly, the expression of S100A10 demonstrated a positive relationship with Kahalidef and Irofulven ($P<0.05$). Moreover,

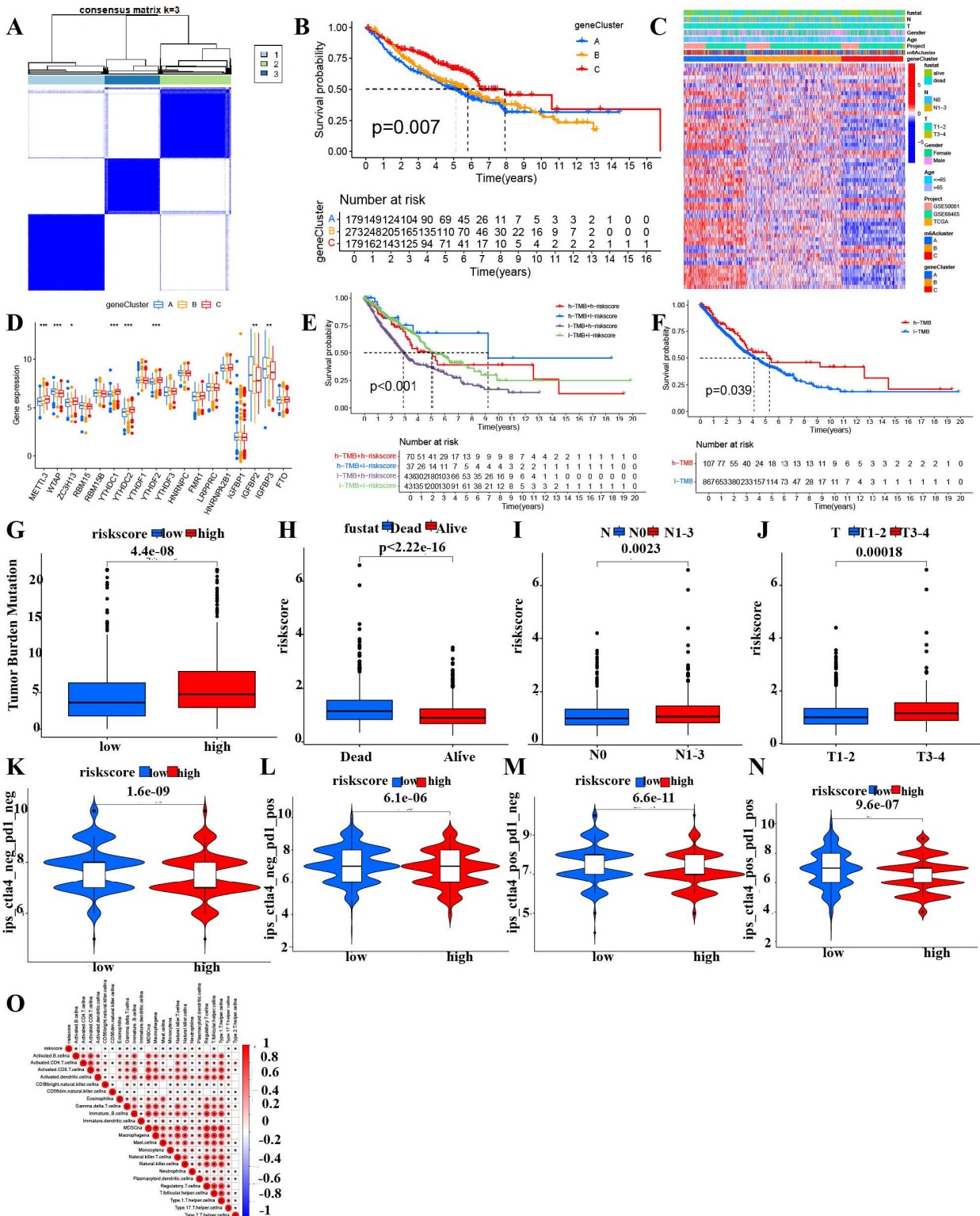

**Fig 2. Integrated genotyping, TMB-riskscore survival, and immune-clinical correlations in NSCLC. A** Gene cluster analysis divided the m6A cluster differential genes into three genotypes. **B** Three gene clusters survival analysis in NSCLC. **C** Heatmap of correlation analysis between m6A clusters, gene cluster expression and clinical pathological characteristics of NSCLC patients. **D** Differential expression analysis of m6A modulators between three

gene clusters. **E-F** TMB combined the riskscore of prediction model survival analysis. **G** Association analysis of TMB and riskscore in NSCLC. **H** Correlation analysis between the survival status of NSCLC patients and risk score. **I-J**. Correlation analysis between the pathological N stages and T stages of NSCLC patients and risk score. **K-N** Violin plot of correlation analysis between IPS and risk score of prediction model of NSCLC patients. Red and blue violin plots indicate the low- and high-risk cohorts. **O** Association analysis between immune infiltration and risk score in NSCLC.

while SLC52A1 expression showed a positive correlation with Fulvestrant, it exhibited a negative correlation with Vinblastine and Depsipeptide ($P<0.05$). Furthermore, the expression levels of KREMEN2 demonstrated a positive correlation with Fulvestrant ($P<0.001$) (Fig 3).

### Clinical and subgroup survival analyses

Survival analysis showed clear differences in clinical outcomes related to age, gender, and tumor characteristics. In the cohort categorized with high-risk scores, patients aged 65 years and older, as well as those under 65 years—encompassing both genders—displayed significantly worse clinical outcomes relative to individuals in the low-risk score cohort ($P<0.001$) (S6A-S6F Fig).Importantly, individuals diagnosed with T3-4 tumors within the low-risk score cohort demonstrated more advantageous clinical outcomes when contrasted with their counterparts in the high-risk score cohort ($P=0.05$) (S6G-S6H Fig).

### Tumor immune microenvironment and m⁶A prognostic genes

The UCSC-Xena database was utilized to download the expression data, immune subtypes, clinical characteristics, and stemness scores for TCGA-LUAD and TCGA-LUSC. Boxplot analysis revealed that S100A10, SCPEP1, STRAP, PAK2, and NPC2 were up-regulated in patients with LUAD and LUSC (S7A-S7B Fig), Correlation analysis indicated that the prognostic m⁶A cluster differential genes exhibited positive correlations with one another (S7C-S7D Fig). Analysis of the immune subtypes of LUAD and LUSC revealed that 19 m⁶A cluster differential genes were distinct in immune subtypes C1, C6, C2, C4, and C3 ($P<0.05$) (S7E-S7F Fig). The expressions of KREMEN2, STRAP, PAK2, ESPL1, TTK, and SKA1 had a positive correlation with the RNAss and DNAss of LUAD ($P<0.05$). The levels of expression for S100A10, SCPEP1, COL4A3, PCDH7, PAK2, CYP4B1, PRRX2, NPC2, and CCR7 exhibited a positive correlation with the estimated score of LUAD ($P<0.05$) (S8A Fig). In patients diagnosed with LUSC, a significant positive correlation was observed between the expressions of SLC52A1, GSTM2, STRAP, PAK2, ESPL1, TTK, and SKA1 with the RNA based stemness scores (RNAss), with a statistical significance of ($P<0.05$). Conversely, the expressions of S100A1, IGFBP1, COL4A3, COL4A4, NPC2, and CCR7 exhibited a significant negative correlation with the DNA methylation-based stemness score (DNAss) among LUSC patients ($P<0.05$). Additionally, a positive association was identified between the expressions of COL4A3, PCDH7, CYP4B1, COL4A1, NPC2, and CCR7 and the stromal score within this patient cohort ($P<0.05$). Notably, the expressions of SLC52A1, GSTM2, STRAP, PAK2, ESPL1, TTK, and SKA1 were inversely correlated with both the immune score and the estimate score of LUSC patients, achieving statistical significance ($P<0.05$) (S8B Fig).

### Gene expression and clinical outcomes

The expression of ESPL1 was statistically distinct between pathological M1 and M0 stages of LUAD patients ($P<0.05$) (S9A Fig). Expressions of SKA1 and CCR7 were statistically different between pathological M1 and M0 stages of LUSC patients (S9E Fig). In LUAD patients, the expressions of S100A10, IGFBP1, GSTM2, STRAP, CYP4B1, PRRX2, TTK, and SKA1 differed between pathological N1-3 and N0 stages ($P<0.05$) (S9B Fig). For LUSC patients, the expressions of CYP4B1, ESPL1, SKA1, and NPC2 differed between pathological N0 and N1-3 stages ($P<0.05$) (S9F Fig). The expressions of SCPEP1, COL4A3, CYPB1, MTUS1, COL4A4, and CCR7 differed in pathological T1-4 stages of LUSC patients ($P<0.05$), while there was no differential expression of the m⁶A cluster differential genes in pathological T1-T4 stages

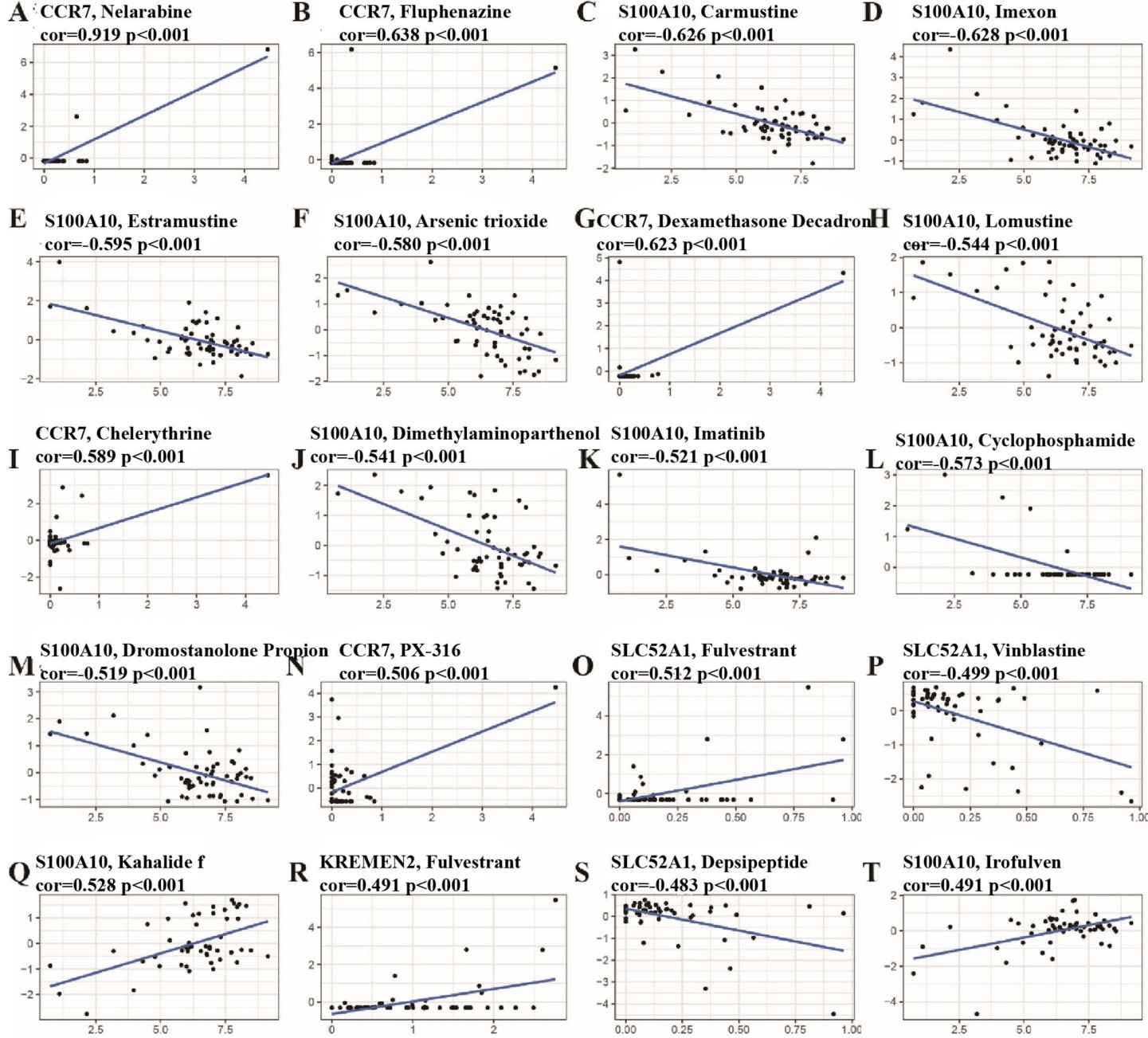

**Fig 3. Correlation between differential genes of m⁶A group in CellMiner and susceptibility to anti-tumor drugs in NSCLC. A-T** The differential expression of genes within the initial 20 m⁶A groups exhibited a noteworthy correlation with various anti-tumor agents. Specifically, the expression levels of CCR7 demonstrated a positive correlation with Nelarabine, Fluphenazine, Dexamethasone Decadron, Chelerythrine, and PX-316. Conversely, the expression of S100A10 revealed a negative correlation with Carmustine, Imexon, Estramustine, Arsenic trioxide, Lomustine, Dimethylaminoparthenol, Imatinib, Cyclophosphamide, and Dromostanolone Propior, while showing a positive correlation with Kahalide f and Irofulven. Additionally, the expression of SLC52A1 was positively correlated with Fulvestrant, yet negatively associated with Vinblastine and Depsipeptide.

of LUAD patients (S9G, S9C Fig). In LUAD patients, the expressions of SCPEP1, GSTM2, MTUS1, and SKA1 differed between pathological stages I-IV ($P<0.05$) (S9D Fig). For LUSC patients, the expressions of KREMEN2, CYP4B1, ESPL1, SKA1, and TTK were statistically different among pathological stages I-IV ($P<0.05$) (S9H Fig). The findings from the survival analysis indicated that elevated levels of ESPL1, IGFBP1, KREMEN2, PAK2, PCDH7, PRRX2, S100A10, STRAP, and TTK in LUAD correlate with unfavorable prognostic outcomes ($P<0.05$) (S10C, S10E, S10F, S10I, S10J, S10K, S10L, S10O, S10P Fig). In LUSC patients, higher expression of CYP4B1 was linked to poor prognosis ($P=0.020$) (S10Q Fig). The elevated expression levels of CCR7, CYP4B1, GSTM2, MTUS1, NPC2, SCPEP1, and SLC52A1 in patients diagnosed with LUAD were associated with an improved prognosis ($P<0.05$) (S10A, S10B, S10D, S10G, S10H, S10M, and S10N Fig). Elevated expression of SLC52A1 in LUSC patients was associated with better clinical outcomes ($P=0.004$) (S10R Fig). The study found notable differences in the levels of 19 m6A cluster genes across LUAD, LUSC, and normal lung tissues ($P<0.05$) (S11-S12 Figs).

**Validation and RT-qPCR confirmation of m6A prognostic genes in NSCLC tissues and cell lines**

The HPA database validated the prognostic significance of differential gene expression in the m6A cluster within NSCLC tissues. This analysis demonstrated statistically significant variations in the expression levels of several genes, including S100A10, SCPEP1, KREMEN2, GSTM2, STRAP, PCDH7, PAK2, ESPL1, TTK, NPC2, and SKA1, when comparing LUAD and LUSC tissues. (see S13 Fig). A number of genes-CCR7, COL4A3, COL4A4, CYP4B1, ESPL1, IGFBP1, MTUS1, PRRX2, SCPEP1, SKA1, SLC52A1, and TTK showed significant expression differences between A549 and Beas-2B cells ($P<0.05$). Moving from this LUAD comparison to additional LUAD lines, ESPL1, GSTM2, KREMEN2, NPC2, PRRX2, SCPEP1, SKA1, SLC52A1, STRAP, and TTK were significantly upregulated in Beas-2B relative to H1299 ($P<0.05$). Like-wise, when comparing H1975 to Beas-2B, ESPL1, GSTM2, MTUS1, NPC2, PCDH7, S100A10, and SCPEP1 displayed significant expression differences ($P<0.05$) (S14 Fig). Shifting focus to LUSC cell lines, H226 exhibited significantly higher COL4A3, COL4A4, MTUS1, NPC2, PAK2, S100A10, and STRAP expression than Beas-2B ($P<0.05$). Conversely, CCR7, GSTM2, PCDH7, SCPEP1, SKA1, and SLC52A1 were expressed at lower levels in the LUSC panel (H1703, H520, H226) compared with Beas-2B ($P<0.05$) (S15 Fig).

**Validation of m6A prognostic genes using scRNA-seq dataset of 20 NSCLC samples from the GEO dataset (GSE119911)**

Fig 4A demonstrated an absence of a notable correlation between the depth of sequencing and both the mitochon-drial gene content and the gene count per sample. Notably, 1500 genes showed high coefficients of variation. The top ten genes with the largest variations included SFTPC, TPSB2, JCHAIN, TPSAB1, SCGB1A1, SCGB3A1, BPIFB1, BPIFA1, GNLY, and COL3A1 (Fig 4B). Fig 4C illustrated the relationship among the number of gene sequences obtained per sample, sequencing depth, and the percentage of mitochondrial genes. PCA-based dimensionality reduction identified different cell types in NSCLC tissues (Fig 4D). Fig 8E also displayed 20 signature genes spanning PC 1–4. Fig 4F to 4H provided PCA heat maps and $p$-value distributions for each principal component. Utilizing 20 PCs, UMAP clustering identified 22 cell clusters (Fig 4I). Annotations identified seven cell types: Monocyte, B cells, T cells, epithelial cells, iPS cells, Macrophage, and tissue stem cells (Fig 5A). The bubble chart illustrated specific gene expression patterns. SCPEP1 was upregulated in macrophages, epithelial cells, and monocytes; STRAP was elevated in iPS cells; PAK2 was found in macrophages, monocytes, and iPS cells; MTUS1 was expressed in tissue stem and epithelial cells; and NPC2 was prominent in epithelial cells (Fig 5B). Fig 5C displays how gene expression linked to prognostic m6A clusters varies among different cell types. The violin plots indicate an upregulation of STRAP, PAK2, and NPC2 in multiple cell types. In contrast, SCPEP1, MTUS1, PCDH7, and SKA1 exhibit differential expression across distinct cell types (Fig 6).

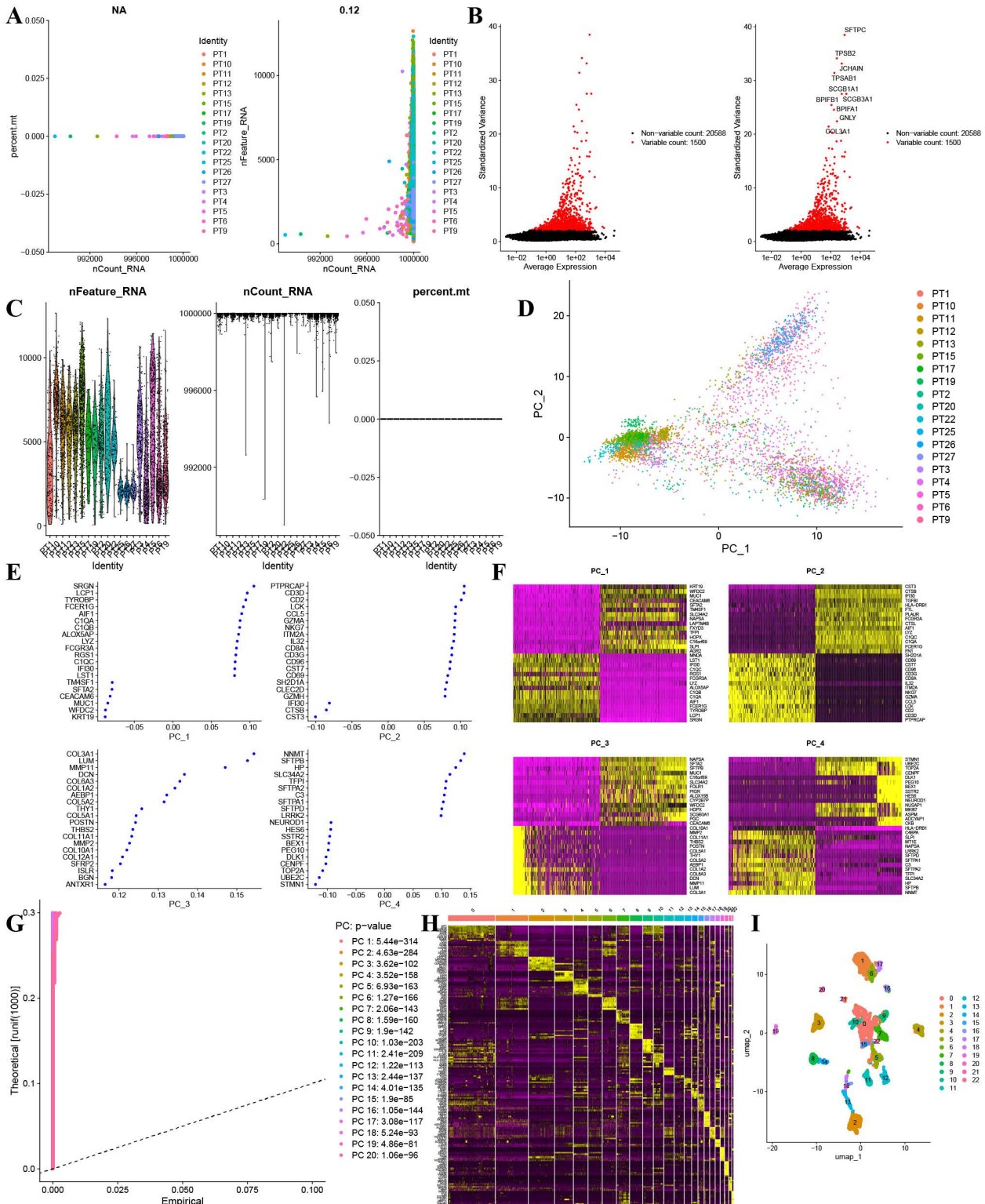

**Fig 4. Normalization, dimensionality reduction, and quality control of cell trajectories were performed on scRNA-seq data. A**. Correlation analyses were conducted on mitochondrial gene sequences alongside total intracellular sequences in relation to sequencing depth. **B.** A comprehensive examination encompassing 20,588 genes revealed that 1,500 genes displayed significant intercellular variability, while 19,088 genes demonstrated

minimal variation. Furthermore, 10 genes were identified as exhibiting the highest levels of variation, as depicted in the accompanying figure. **C**. Following the application of quality control normalization procedures, a total of 5,406 cells were harvested for subsequent analysis. **D**. PCA utilizing scRNA-seq data. **E** Emphasizing the twenty genes that displayed the most pronounced variations in each PCA. **F**. A heatmap illustrating the expression levels of the top 20 genes that exhibit the most significant differences in each PCA has been generated. **G**. The most prominent 20 gene expression heat curves in each PCA. **H**. 5406 single cells were divided into 22 cohorts, of which 6386 marker genes were displayed on the heatmap. **I**. A total of 22 clusters were visualized using UMAP.

## Network pharmacology reveals IGFBP1 as a m⁶A target of Pueraria in NSCLC

We integrated data from the GeneCards, CTD, TTD, and OMIM databases to obtain 7,333 NSCLC-related genes (Fig 7A), and based on Herb2.0, screened for active components of Pueraria (such as 7,8,4'-trihydroxyisoflavone and daidzein), resulting in 846 target points for Pueraria after pharmacokinetic screening via SwissADME and target prediction through SwissTargetPrediction. Then, we intersected the 366 target genes of Pueraria with the NSCLC-related genes to obtain 249 intersecting genes (Fig 7B). GO enrichment analysis of the intersecting genes showed that these genes are mainly enriched in protein serine/threonine kinase activity and positive regulation of the MAPK cascade (Fig 7C-7E), while KEGG pathway enrichment analysis indicated that these genes are primarily enriched in the PI3K/AKT signaling pathway (Fig 7F and 7G). Subsequently, we further constructed a regulatory network of Pueraria-effective drug ingredients-target genes-KEGG pathways-NSCLC (Fig 7H). We then intersected the target genes of Pueraria with NSCLC-related genes and the previously screened prognostic m⁶A-related gene cluster, ultimately obtaining IGFBP1 (Fig 7I).

## Validation of IGFBP1 expression in NSCLC tissues using IHC

Fig 8A and 8B present IHC staining of IGFBP1 in two LUAD tissues. The overall staining appears to be weak to moderate across the tissue. The histological arrangement shows tightly packed tumor cells with discernible nuclei and cytoplasm. IHC staining of IGFBP1 in LUSC tissue demonstrating a stronger signal compared to LUAD samples. The image shows notable IGFBP1 expression within the tumor cells, particularly at the outer layers. The inset shows increased IGFBP1 staining within tumor cells arranged in concentric layers (Fig 8C). Another representation of IGFBP1 staining in LUSC tissue. Similar to (C), there is moderate expression throughout the tumor cells. The inset highlights the presence of more intense staining for IGFBP1 within the cytoplasm of the squamous cells (Fig 8D). The IHC staining of IGFBP1 in LUAD and LUSC tissues was consistent with the expression of IGFBP1 in TCGA tissues (S11 and S12 Figs).

## Molecular docking and MD simulations validation the stability binding of 7, 8, 4'-trihydroxyisoflavone-IGFBP1 and Genistein-IGFBP1

We then analyzed the molecular docking of IGFBP1 with two effective components of Pueraria through the CB-dock2 website, showing that IGFBP1 has a stable binding with 7,8,4'-trihydroxyisoflavone and Genistein from Pueraria (Fig 9A-9B). We utilized the root mean square deviation (RMSD) as an indicator to assess the establishment of a stable state within the simulated system. An RMSD measurement of 1 nm or less indicates the relative stability of the protein-ligand interaction under physiological circumstances. The RMSD for the 7,8,4'-trihydroxyisoflavone-IGFBP1 complex displayed a gradual increment, achieving a plateau after 40 ns, ultimately stabilizing at 1.2 nm (Fig 10A). In a similar manner, the RMSD for the Genistein-IGFBP1 complex also reached stabilization at 1.25 nm (Fig 10E). To further assess the compactness of the receptor-ligand binding, we investigated the radius of gyration (Rg). The Rg values for the 7,8,4'-trihydroxyisoflavone-IGFBP1 reached stability at 2.2 nm and the Genistein-IGFBP1 complexes reached stability at 2.25 nm (Fig 10B and 10F). These findings suggest an increase in the overall compactness of the complexes, indicating that the system was progressing towards a stable configuration. Another crucial aspect that reflects protein stability and folding is the solvent-accessible surface area (SASA). Our analysis indicated that the SASA value for the 7,8,4'-trihydroxyisoflavone-IGFBP1 complex remained stable, decreasing from 200 nm² to 175 nm² over time (Fig 10C). The SASA for the Genistein-IGFBP1

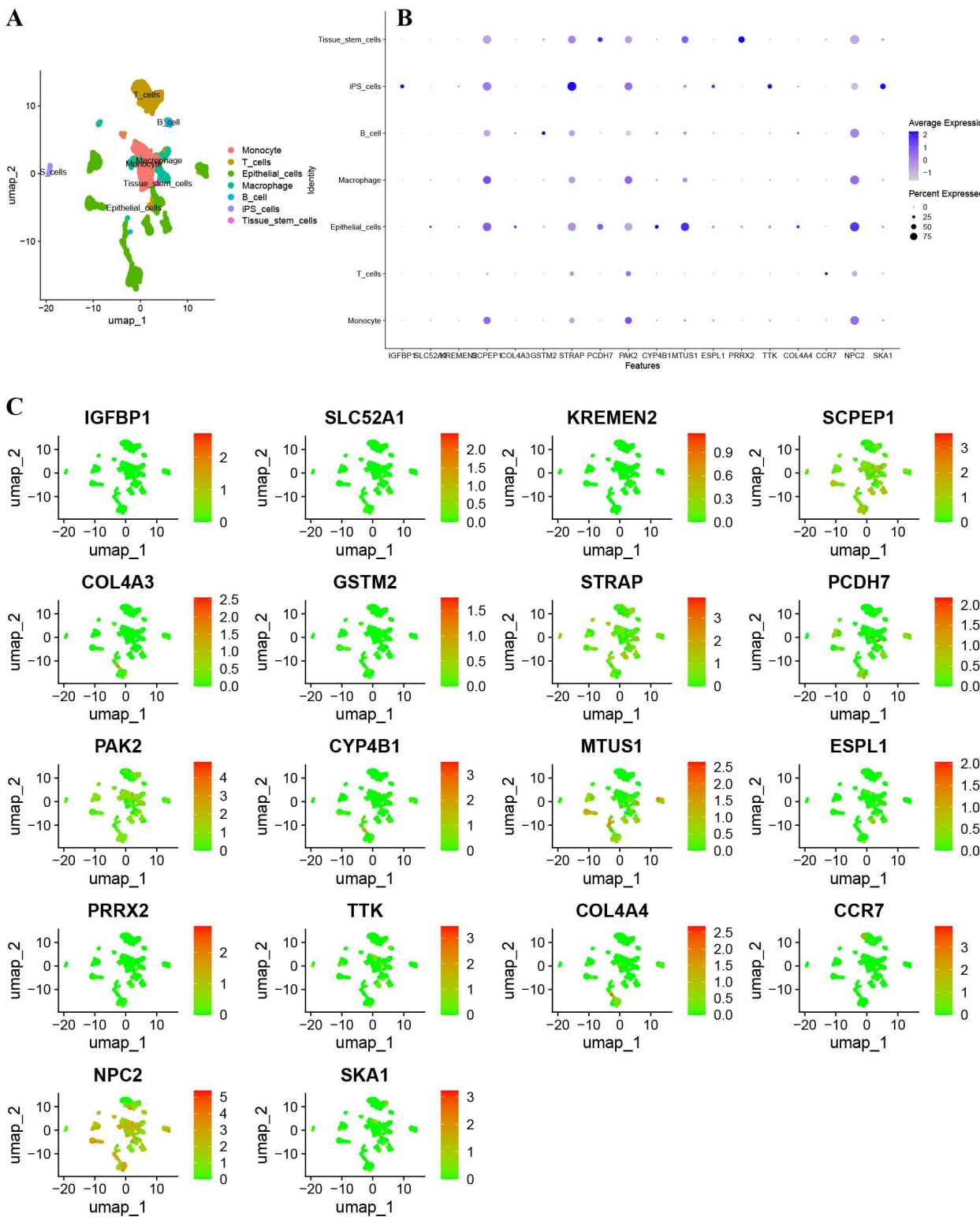

**Fig 5. Visual analysis of prognostic m⁶A cluster differential genes. A** Display of visualization results after cell annotation **B** Bubble chart of prognostics m⁶A cluster differential genes in each cluster. **C** Scatter plot of prognostic m⁶A cluster differential genes in each cluster.

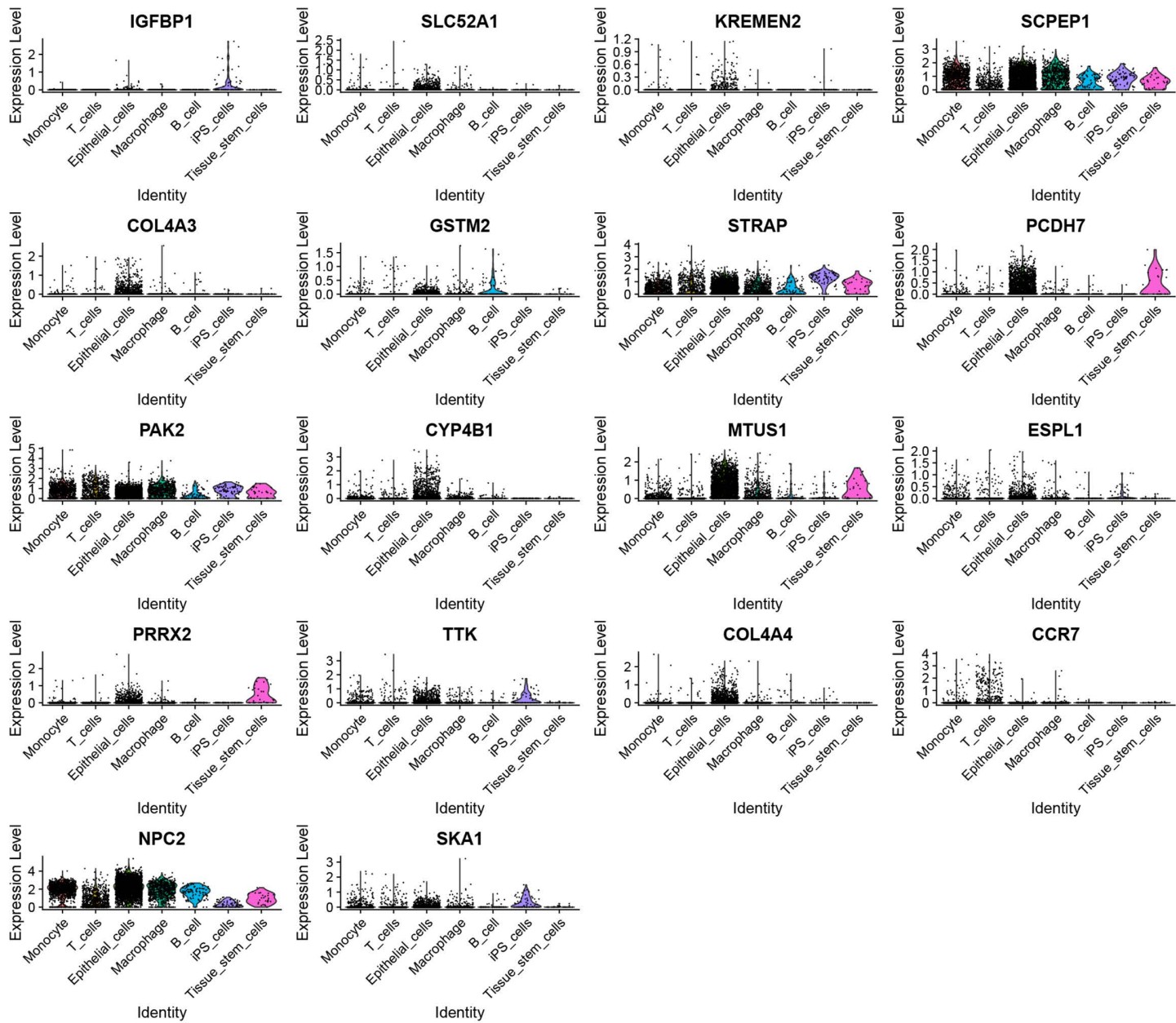

**Fig 6. Violin plots of prognostic m⁶A cluster differential genes showing differences in different cells based on scRNA-seq dataset.** The genes SCPEP1, STRAP, PAK2, and NPC2 demonstrated elevated expression levels across various cell types, including monocytes, T cells, epithelial cells, macrophages, B cells, induced pluripotent stem (iPS) cells, and tissue stem cells. Notably, the genes SLC52A1, KREMEN2, COL4A3, GSTM2, PCDH7, CYP4B1, MTUS1, ESPL1, PRRX2, TTK, COL4A4, and SKA1 showed significant expression specifically in epithelial cells. Additionally, PCDH7, MTUS1, and PRRX2 were found to be prominently expressed in tissue stem cells.

complex diminished from 200 nm² to 170 nm², achieving equilibrium by 40 ns (Fig 10G). Furthermore, the number of hydrogen bonds serves as an indicator of the strength of the protein-ligand interaction; both the 7,8,4'-trihydroxyisoflavone-IGFBP1 and Genistein-IGFBP1 complexes exhibited a consistent density and strength of hydrogen bonds throughout the duration of the simulation (Fig 10D and 10H).

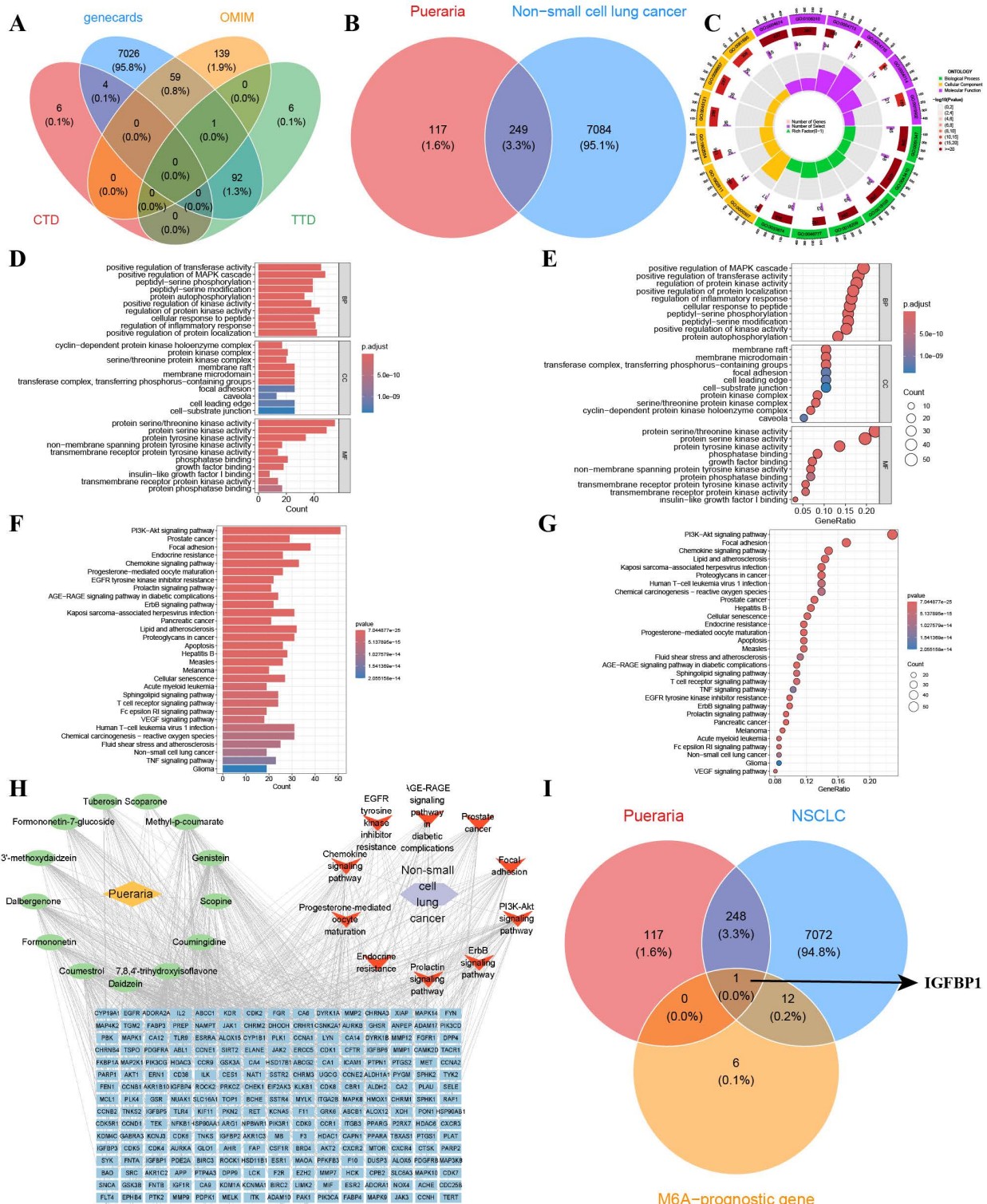

**Fig 7. Network Pharmacology unveil IGFBP1 as a m6A epigenetic target of Pueraria in NSCLC. A.** Downloaded and combined NSCLC-related genes from Genecards, CTD, TTD, and OMIM databases. **B.** The intersection of genes linked to Pueraria and NSCLC. **C-E.** GO analysis of intersection genes. **F-G** KEGG analysis of intersection genes. **H** Construct a regulatory network of Pueraria -drug ingredients-target genes-KEGG pathways-NSCLC. **I** Intersect target genes of Pueraria with NSCLC and prognostic m6A-cluster differential genes from screening.

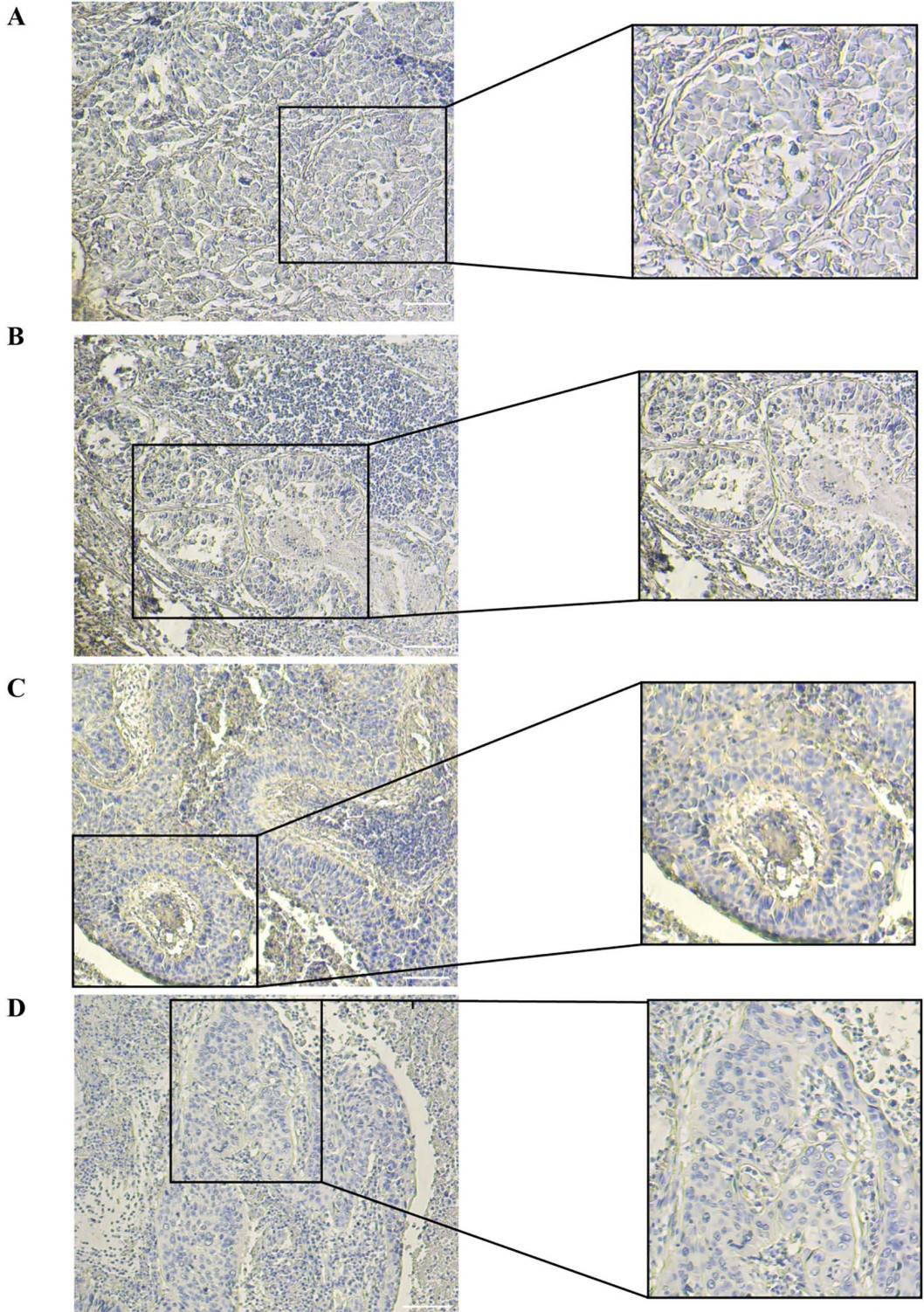

**Fig 8. Immunohistochemical Staining of IGFBP1 in LUAD and LUSC Tissues.** IGFBP1 Immunohistochemistry: Representative staining in LUAD (A, B) and LUSC (C, D) tissues. IGFBP1 showed positive expression in LUAD and LUSC tissues.

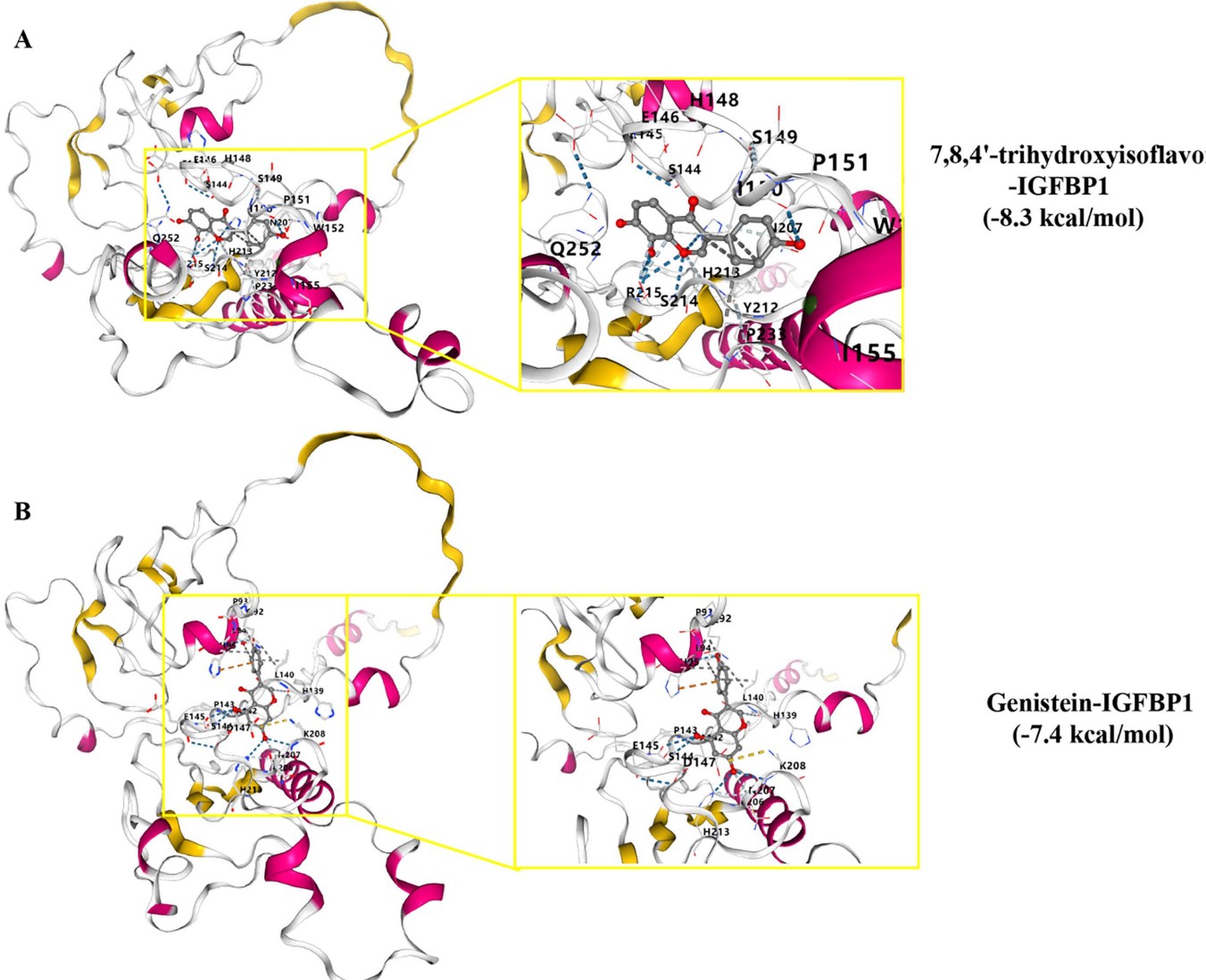

**Fig 9. Molecular docking analysis. A** Molecular docking diagram of 7,8,4'-trihydroxyisoflavone, an active ingredient of Pueraria, with IGFBP1. The 3D structure of the ligand was obtained from PubChem, and the protein structure from the AlphaFold DB. **B** Molecular docking diagram of genistein, an active ingredient of Pueraria, with IGFBP1. The 3D structure of the ligand was obtained from PubChem, and the protein structure from the AlphaFold DB.

## Discussion

M⁶A RNA methylation is crucial for tumor immunotherapy, with binding proteins affecting malignancy [29]. Key m⁶A regulators impact the tumor microenvironment (TME) [30], especially in early-stage LUAD [31]. Our study shows m⁶A regulators can identify TME traits, and a differential gene framework may reveal biomarkers for NSCLC immunotherapy.

Smoking links to NSCLC subtypes LUAD and LUSC, often promoting metastasis by altering the TME and activating metastatic genes [32]. Recent research highlights m⁶A modifications role in NSCLC development, such as smoking-induced M2 TAM formation via circular EML4 in EVs, mediated by ALKBH5-regulated m⁶A of SOCS2 [33]. Smoking influences subtype prevalence and molecular changes, important for understanding and targeting lung cancer, especially in smokers.

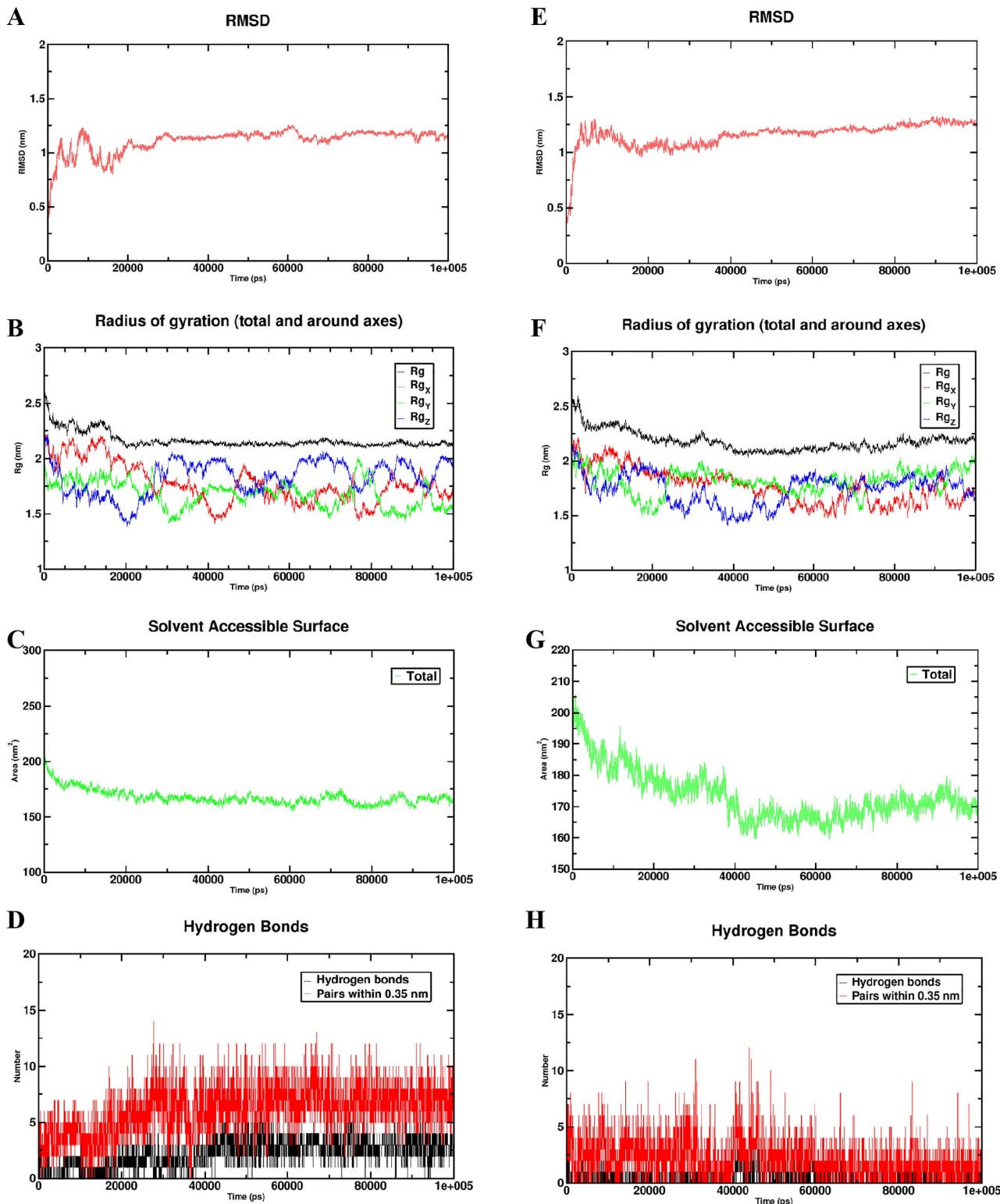

**Fig 10. Molecular dynamics (MD) simulations analysis of the binding and stability affinity of the 7,8,4'-trihydroxyisoflavone-IGFBP1 and the Genistein-IGFBP1 complex.** **A** RMSD values of the 7,8,4'-trihydroxyisoflavone-IGFBP1 complex over time. **B** Rg values of the 7,8,4'-trihydroxyisoflavone-IGFBP1 complex. **C** SASA values of the 7,8,4'-trihydroxyisoflavone-IGFBP1 complex. **D** Number of hydrogen bonds in the

7,8,4'-trihydroxyisoflavone-IGFBP1 complex, highlighting the strengthen of the protein-ligand interaction. **E** RMSD values of the Genistein-IGFBP1 complex over time. **F** Rg values of the Genistein-IGFBP1 complex. **G** SASA values of the Genistein-IGFBP1 complex. **H** Number of hydrogen bonds in the Genistein-IGFBP1 complex, highlighting the strengthen of the protein-ligand interaction.

Our analysis identified 19 differential genes (S100A10, IGFBP1, SLC52A1, KREMEN2, SCPEP1, COL4A3, GSTM2, STRAP, PCDH7, PAK2, CYP4B1, MTUS1, ESPL1, PRRX2, TTK, COL4A4, CCR7, NPC2, SKA1) related to immune subtypes, clinical characteristics, stemness, and drug sensitivity. Several of these genes have mechanistic or translational relevance: NPC2 supports immune-related stratification and therapeutic sensitivity through PI3K/AKT regulation [34]. SLC52A1 likely reflects immunometabolic states associated with stable drug-resistant cell populations [35]. Importantly, STRAP has been implicated in immune-infiltrated risk phenotypes, supporting its role in shaping the tumor immune microenvironment [36]. The translational relevance is further reinforced by evidence for PAK2 in early detection and treatment response prediction [37], and PCDH7 in malignant epithelial–immune interactions [38]. We acknowledge that immune interactions in the TME are complex and only partly explained by m6A-related expression patterns. Mechanistic studies indicate that m6A regulators can modulate key immune signaling pathways. For example, ALKBH5-mediated m6A activates JAK2/p-STAT3, enhancing NSCLC progression and influencing anti-PD-L1 therapy response [39]. The immune microenvironment is a key aspect, but tumor behavior is also affected by genetic alterations and cellular interactions across m6A clusters [40,41]. The genomic traits of m6A clusters may influence cell subsets and cytokine secretion, impacting immune responses and highlighting the link between immune cell infiltration and these clusters. We propose a focused pathway for clinical translation. First, rigorously evaluate lead compounds, such as Pueraria isoflavonoids, for target engagement, specificity, toxicity, and efficacy. Next, integrate single-cell and spatial transcriptomics with profiling of immune cell markers to map the effects of m6A dysregulation. Finally, translate validated signatures and compounds into clinical assays (qPCR/NGS panels or IHC) and early-phase trial frameworks for rapid evaluation. Future studies will include more immune markers to clarify m6A cluster differences' biological significance.

We developed a prognostic network for NSCLC, identifying IGFBP1 and HNRNPC as risk factors, aligning with previous findings that HNRNPC is a prognostic marker in LUAD [42]. NSCLC was categorized into three clusters based on m6A regulators, revealing pathway activations such as JAK-STAT and Hedgehog. Mechanistic links between m6A regulators and these pathways have been reported: YTHDF3 has been shown to potentiate JAK/STAT antiviral responses by promoting translation of signaling components [43]. In our cohort, m6A gene clusters were associated with differences in immune cell composition and immunophenoscore, with significant variations in B cells, T cells, and dendritic cells across clusters. While these associations are correlative, prior functional studies provide mechanistic plausibility: The immune cell infiltration pattern is one aspect of m6A cluster differences; m6A modification also affects the TME by regulating gene expression, metabolism, and signaling pathways [44–46], and m6A modulators as potential biomarkers and therapeutic targets in esophageal cancer immunotherapy [47]. Clinical correlations between immune-rich ("hot") tumors and improved response to immune checkpoint blockade have been documented across cancer types [48]. These findings align with studies emphasizing m6A's role as a biomarker and therapeutic target, especially in "hot" tumors with high immune infiltration, which respond better to immunotherapy.

In our cohort, The low-risk group with high IPS and CTLA-4/PD1 positivity responds better to immunotherapy than the high-risk group ($P = 9.6e\text{-}07$). Furthermore, M6A cluster genes further enhance this response, highlighting their role in immune modulation and cancer treatment, especially in hepatocellular carcinoma [49]. Studies link m6A signatures in melanoma to immune infiltration as potential biomarkers [50]. Research on pancreatic adenocarcinoma explored m6A methylation modulators and TME infiltration [51].These studies emphasize m6A modulators' role in tumor immune infiltration and their potential as cancer immunotherapy targets.

Moreover, integrating network pharmacology, we connected traditional herbal compounds to modern oncology. By curating 7,333 NSCLC-associated genes, 16 active compounds and 366 bioactive targets of Pueraria, we identified 249 hub genes involved in kinase activity and MAPK/PI3K-AKT pathways—key for NSCLC progression and resistance. IGFBP1 emerged as a core target, linking m6A regulation, tumor microenvironment, drug sensitivity, and immune infiltration. Previous studies on m6A in NSCLC have largely focused on retrospective bioinformatics analyses that construct m6A-related gene prognostic signatures for risk stratification [52–54], without delving deeper into the relationship between these prognostic genes and monomeric compounds in traditional Chinese medicine. This approach highlights the potential of repurposing phytochemicals to target RNA-modification networks and modernize herbal medicine; however, potential confounders (patient demographics, smoking status, treatment history, and batch effects in public datasets) may influence these associations and were considered in our analyses. Molecular docking validated the therapeutic potential of Pueraria's isoflavonoids, especially 7,8,4'-trihydroxyisoflavone and genistein, which showed high-affinity binding to IGFBP1. These interactions connect m6A dysregulation with herbal pharmacology, positioning IGFBP1 as a link for epigenetically guided NSCLC therapy. By integrating multi-omics insights with single-cell validation, this work offers a blueprint for targeting RNA modification hubs through natural compounds, advancing precision oncology and herbal medicine.

While the dependence on computational methodologies and the validation of tissues and cells presents certain limitations, future investigations should encompass more comprehensive functional assays. Furthermore, it is imperative to pursue broader patient validations to substantiate these findings. Although molecular docking and dynamics indicate stable interactions, further exploration into the pharmacokinetics and safety profile of Pueraria compounds is warranted through subsequent preclinical studies.

## Conclusions

This study reveals a novel RNA epigenetic mechanism underlying Pueraria's anti-tumor effects in NSCLC, identifying IGFBP1 as a key therapeutic target. The integration of multi-omics and network pharmacology offers a promising approach for modernizing traditional Chinese medicine and suggests that m6A-related genes may serve as important biomarkers to guide immunotherapy response and personalized treatment strategies in NSCLC.

## Supporting information

**S1 Fig. Copy number alterations, somatic mutation burden, differential expression of 23 m6A associated modulators in NSCLC.** A The maftools of somatic mutation burden frequency of 23 m6A-associated modulators in NSCLC. B The copy number alteration frequency of 23 m6A-associated modulators in NSCLC. C The copy number circle graph of 23 m6A-associated modulators in 23 human chromosomes. D The differential expression level of 23 m6A-associated modulators in NSCLC and normal tissues.
(TIF)

**S2 Fig. Survival analysis of m6A regulated modulators in 1622 NSCLC patients.** The red and blue lines reveal m6A regulators with high and low expression, respectively. A, B, C, D, F, G, I, J, M Survival analysis showed that high expression of FMR1, HNRNPA2B1, HNRNPC, IGFBP1, IGFBP3, LRPPRC, RBM15, WTAP, ZC3H13 have a poor prognosis compared with low expression of them. E, H, K, L Survival analysis showed that low expression of IGFBP2, METTL3, YTHDC2, YTHDF2 have a poor prognosis compared with high expression of them.
(TIF)

**S3 Fig. Survival analysis of m6A regulated modulators in LUAD patients.** A, I, J Survival analysis showed that low expression of FMR1, YTHDF2, FTO have a poor prognosis compared with the high expression of them. B-H Survival analysis showed that high expression of HNRNPA2B1, HNRNPC, IGFBP1, IGFBP3, LRPPRC, RBM15B, RBM15 have a poor prognosis compared with the low expression of them.
(TIF)

**S4 Fig. Survival analysis of m⁶A regulated modulators in LUSC patients.** A, B, C, E, F, G, H, I Survival analysis showed that low expression of FMR1, HNRNPA2B1, HNRNPC, IGFBP2, LRPPRC, METTL3, WTAP, RBM15 have a poor prognosis compared with high expression of them. D, J, K Survival analysis showed that high expression of IGFBP1, YTHDF1, FTO have a poor prognosis compared with low expression of them.
(TIF)

**S5 Fig. Clinical subgroup survival analysis.** A-H Survival analysis focusing on clinical attributes among low-risk and high-risk groups of patients diagnosed with NSCLC. Patients with NSCLC who possess a high-risk score, particularly those aged over 65 or under 65, regardless of gender, as well as those classified with nodal involvement (N0 and N1-3) and tumor stages (T1-2 and T3-4), exhibit a poorer prognosis compared to their low-risk score counterparts. The red and blue lines represent cohorts categorized as high risk and low risk, respectively.
(TIF)

**S6 Fig. Validation the expression of m⁶A-related regulators using immunohistochemistry in HPA database.** The protein expressions of ALKBH5, FMR1, FTO, HNRNPA2B1, HNRNPC, LRPPRC, RBM15, RBMX, YTHDC1, YTHDC2, YTHDF2, and ZC3H13 were strongly positive in LUAD and LUSC tissues, while the protein expressions of METTL14, METTL16, RBM15B, VIRMA, and WTAP were weakly positive in LUAD and LUSC tissues.
(TIF)

**S7 Fig. Expression, correlation, immune subtype analysis of prognostic m⁶A cluster differential genes.** A-B Boxplot of the 19 prognostic m⁶A cluster differential genes in LUAD and LUSC tissues from UCSC-Xena. C-D Correlation analysis of the expression of the 19 prognostic m⁶A cluster differential genes in LUAD and LUSC. E-F The relationship between the expression of 19 m⁶A cluster differential genes and different immune subtypes.
(TIF)

**S8 Fig. Association analysis between m⁶A cluster differential gene expression and stemness score, immune microenvironment in NSCLC.** A The relationship between m⁶A cluster differential gene expression and stemness score, immune microenvironment in LUAD. B Association analysis of m⁶A cluster differential gene expression and stemness score, immune microenvironment in LUSC.
(TIF)

**S9 Fig. Correlation analysis between 19 prognostic m⁶A cluster differential genes and clinical characteristics of NSCLC.** A-D The relationship between m⁶A cluster differential gene expression and clinical pathological features of LUAD patients. E-H Association analysis of m⁶A cluster differential genes expression and clinical pathological features of LUSC patients.
(TIF)

**S10 Fig. Survival analysis of prognostic m⁶A cluster differential genes in NSCLC.** A-P Survival analysis of m⁶A cluster differential genes in LUAD. High expression of CCR7, CYP4B1, GSTM2, MTUS1, NPC2, SCPEP1, SLC52A1 had a better clinical outcome than low expression of them in LUAD patients. Low expression of ESPL1, IGFBP1, KREMEN2, PAK2, PCDH7, PRRX2, S100A10, STRAP, TTK had a better clinical outcome than high expression of them in LUAD patients. Q-R Survival analysis of m⁶A cluster differential genes in LUSC. High expression of CYP4B1 had a poorer prognosis than low expression of CYP4B1 in LUSC patients. High expression of SLC52A1 had a better clinical outcome than low expression of SLC52A1 in LUSC patients.
(TIF)

**S11 Fig. Differential expression of 19 prognostic m⁶A cluster differential genes in normal and LUAD tissues.** A-S Red and blue boxplots indicate LUAD and normal tissues in the TCGA database. CCR7, ESPL1, KREMEN2, NPC2,

PAK2, PCDH7, PRRX2, S100A10, SCPEP1, SKA1, SLC52A1, STRAP, TTK were highly expressed in LUAD tissues. COL4A3, COL4A4, CYP4B1 were down-expressed in LUAD tissues. * Represents $P<0.05$, ** represents $P<0.01$, *** represents $P<0.001$, **** represents $P<0.0001$.
(TIF)

**S12 Fig. Differential analysis of 19 prognostic m⁶A cluster differential genes in LUSC and adjacent non-LUSC tissues.** A-S Red and blue boxplots indicate LUSC and non-LUSC tissues from the TCGA database. CCR7, COL4A3, COL4A4, CYP4B1, MTUS1, NPC2, S100A10 were highly expressed in non-LUSC tissues. ESPL1, GSTM2, IGFBP1, KREMEN2, PAK2, PCDH7, PRRX2, SCPEP1, SKA1, SLC52A1, STRAP, TTK were highly expressed in LUSC tissues. * Represents $P<0.05$, ** represents $P<0.01$, *** represents $P<0.001$, **** represents $P<0.0001$.
(TIF)

**S13 Fig. Validation m⁶A cluster differential gene expression in HPA database.** The protein expressions of S100A10, GSTM2, STRAP, PCDH7, PAK2, ESPL1, TTK, SKA1 were strongly positive in LUAD and LUSC tissues, The protein expression of NPC2 was strongly positive in LUAD tissues and negative in LUSC tissues.
(TIF)

**S14 Fig. Validation 19 m⁶A cluster differential gene expression in LUAD and Beas-2B cell lines.** A-S The expressions of CCR7, COL4A3, COL4A4, CYP4B1, IGFBP1, MTUS1, SCPEP1, SLC52A1 were highly expressed in A549 cell lines. The expression of ESPL1, GSTM2, KREMEN2, NPC2, PAK2, PCDH7, PRRX2, SCPEP1, SKA1, STRAP, TTK were highly expressed in H1299 cell lines. The expressions of ESPL1, GSTM2, MTUS1, NPC2, PCDH7, S100A10, SCPEP1, SKA1 were highly expressed in H1975 cell lines. * Represents $P<0.05$, ** represents $P<0.01$, *** represents $P<0.001$, **** represents $P<0.0001$.
(TIF)

**S15 Fig. Validation 19 m⁶A cluster differential gene expression in LUSC and Beas-2B cell lines.** A-S The expressions of CCR7, COL4A3, ESPL1, GSTM2, IGFBP1, MTUS1, NPC2, PCDH7, S100A10, SCPEP1, SKA1, SLC52A1 were down-expressed in H1703 celll lines. The expressions of PRRX2, STRAP were highly expressed in H1703 cell lines. The expressions of CCR7, COL4A3, CYP4B1, GSTM2, IGFBP1, KREMEN2, MTUS1, NPC2, PCDH7, S100A10, SCPEP1, SKA1, SLC52A1 were down-expressed in H520 cell lines. The expressions of STRAP, TTK were highly expressed in H520 cell lines. The expressions of CCR7, ESPL1, GSTM2, KREMEN2, PCDH7, PRRX2, SCPEP1, SKA1, SCL52A1 were down-expressed in H226 cell lines. The expressions of COL4A3, COL4A4, MTUS1, NPC2, PAK2, S100A10, STRAP were highly expressed in H226 cell lines. * Represents $P<0.05$, ** represents $P<0.01$, *** represents $P<0.001$, **** represents $P<0.0001$.
(TIF)

**S16 Fig. Graphical abstract.** The technology diagram of the whole study.
(TIF)

**S1 Table. The primer sequences of GAPDH and 19 prognostic m⁶A cluster differential genes.**
(XLSX)

**S2 Table. The results of KEGG signaling pathway enrichment differences between m⁶A cluster A and B.**
(XLSX)

**S3 Table. The results of KEGG signaling pathway enrichment differences between m⁶A cluster A and C.**
(XLSX)

**S4 Table. The results of KEGG signaling pathway enrichment differences between m6A cluster B and C.**
(XLSX)

## Author contributions

**Conceptualization:** Rui Li, Yi-Qing Qu.

**Data curation:** Rui Li, Yu-Xin Zhang, Yi-Qing Qu.

**Formal analysis:** Rui Li, Dong-Mei Hu, Wei Zhao, Yi-Qing Qu.

**Funding acquisition:** Yi-Qing Qu.

**Investigation:** Rui Li, Yu-Xin Zhang, Yi-Qing Qu.

**Methodology:** Rui Li, Yong-Li Liu, Wei Zhao, Yi-Qing Qu.

**Project administration:** Yi-Qing Qu.

**Resources:** Rui Li, Dong-Mei Hu, Yong-Li Liu.

**Software:** Rui Li, Dong-Mei Hu, Wei Zhao.

**Supervision:** Yi-Qing Qu.

**Validation:** Yong-Li Liu, Yu-Xin Zhang.

**Visualization:** Yi-Qing Qu.

**Writing – original draft:** Rui Li.

**Writing – review & editing:** Yi-Qing Qu.

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
