## [Decision Letter · Decision Letter 0]

29 Oct 2025

PCOMPBIOL-D-25-01915

Multi-Omics and Network Pharmacology Identify IGFBP1 as an m6A-Epigenetic Target of Pueraria in NSCLC Therapy

PLOS Computational Biology

Dear Dr. Qu,

Thank you for submitting your manuscript to PLOS Computational Biology. After careful consideration, we feel that it has merit but does not fully meet PLOS Computational Biology's publication criteria as it currently stands. Therefore, we invite you to submit a revised version of the manuscript that addresses the points raised during the review process.

Please submit your revised manuscript within 30 days Dec 29 2025 11:59PM. If you will need more time than this to complete your revisions, please reply to this message or contact the journal office at ploscompbiol@plos.org. Please include the following items when submitting your revised manuscript:

We look forward to receiving your revised manuscript.

Kind regards,

Eduardo Jardón-Valadez

Academic Editor

PLOS Computational Biology

Ferhat Ay

Section Editor

PLOS Computational Biology

**Additional Editor Comments:**

Dear Authors,

Please find below the comments and suggestions provided by the invited reviewers to assist you in preparing the revised version of your manuscript. I believe that your study is both valuable and relevant for publication in PLOS Computational Biology.

In your revision, please ensure that the aims and hypotheses are clearly stated, the figure numbering and corresponding references in the main text are consistent, and all acronyms are properly defined. Additionally, I encourage you to discuss your main findings more thoroughly, improve the logical organization of the Results section, and expand the Methods section to include details, eg, in the molecular dynamics (MD) setup, equilibration procedures, production runs, and analysis protocols.

Kind regards

Eduardo Jardon

**Journal Requirements:**

1) Please provide an Author Summary. This should appear in your manuscript between the Abstract (if applicable) and the Introduction, and should be 150-200 words long. The aim should be to make your findings accessible to a wide audience that includes both scientists and non-scientists. Sample summaries can be found on our website under Submission Guidelines:

4) Please provide a detailed Financial Disclosure statement. This is published with the article. It must therefore be completed in full sentences and contain the exact wording you wish to be published.

1) Please clarify all sources of financial support for your study. List the grants, grant numbers, and organizations that funded your study, including funding received from your institution. Please note that suppliers of material support, including research materials, should be recognized in the Acknowledgements section rather than in the Financial Disclosure

2) State the initials, alongside each funding source, of each author to receive each grant. For example: "This work was supported by the National Institutes of Health (####### to AM; ###### to CJ) and the National Science Foundation (###### to AM)."

3) State what role the funders took in the study. If the funders had no role in your study, please state: "The funders had no role in study design, data collection and analysis, decision to publish, or preparation of the manuscript."

4) If any authors received a salary from any of your funders, please state which authors and which funders..

5) Your current Financial Disclosure states, "The author(s) received no specific funding for this work.".

However, your funding information on the submission form indicates receiving fund from National Natural Science Foundation of China.

Please indicate by return email the full and correct funding information for your study and confirm the order in which funding contributions should appear. Please be sure to indicate whether the funders played any role in the study design, data collection and analysis, decision to publish, or preparation of the manuscript.

6) Kindly revise your competing statement in the online submission form to align with the journal's style guidelines: 'The authors declare that there are no competing interests.'

**Reviewers' comments:**

Reviewer's Responses to Questions

**Comments to the Authors:**

Reviewer #1: Dear Authors

The research is informative and shows a strong computational and bioinformatics workflow (clustering, survival, mutation, CNV, immune infiltration, and pathway analysis). It reflects an advanced use of multi-omics data from TCGA and GEO.

However, the text has conceptual and methodological gaps, and the structure lacks clarity for publication in a high-impact journal.

Major Weaknesses

Lack of Clear Objective

The text jumps directly into results (m6A regulators, LUAD/LUSC analysis) without stating the central hypothesis or specific research question.

It’s unclear whether the study aims to identify prognostic markers, immune subtypes, or therapeutic targets.

Missing Methodological Details

No information on dataset size, normalization methods, statistical thresholds (e.g., adjusted p-values, FDR), or the exact bioinformatics pipeline.

The basis for clustering (consensus clustering parameters, number of clusters k, etc.) is not explained.

No Validation

There’s no mention of experimental validation (e.g., RT-qPCR, IHC, or external dataset verification).

Computational results alone may not confirm biological relevance.

Overgeneralized Conclusions

Claims about immune microenvironment regulation and clinical relevance are strong but not supported by mechanistic evidence.

No discussion of potential confounders or limitations.

Figure / Table References Missing

The text refers to results (e.g., subtypes, immune infiltration) but provides no figure or table references, which weakens reproducibility.

Weak Link to Clinical Translation

Although clinical outcomes (survival) are mentioned, there is no clear suggestion on how identified m6A regulators could be applied in diagnosis or treatment.

Language & Structure

Long, complex sentences reduce clarity.

Some transitions between LUAD and LUSC analyses are abrupt.

Recommendations

Add a clear research aim and hypothesis.

Define what biological question the study answers (e.g., “to identify m6A-based prognostic subtypes in NSCLC”).

Expand the Methods section.

Describe all data sources, preprocessing steps, and statistical thresholds.

Include package names, software versions, and cut-off criteria for DEGs or clustering.

Include experimental or external validation.

Validate expression of key m6A genes via RT-qPCR or IHC.

Use an independent GEO dataset for model verification.

Reorganize Results logically.

(a) Subtype identification →

(b) Mutation/CNV landscape →

(c) Immune infiltration →

(d) Prognostic analysis →

(e) Pathway enrichment.

Add visual evidence.

Figures showing Kaplan–Meier survival curves, heatmaps, immune scores, and m6A regulator expression patterns.

Strengthen Discussion.

Compare findings with previous studies on m6A in NSCLC.

Address limitations (e.g., reliance on public data, lack of experimental confirmation).

Improve English clarity.

Shorten sentences, avoid redundancy, and use transition phrases (“In contrast…”, “Furthermore…”).

Provide a concise conclusion.

Emphasize the novelty and practical implications (e.g., “m6A-related genes may guide immunotherapy response in NSCLC”).

Reviewer #2: This is an interesting research manuscript by Rui Li and co-workers on a topic of growing importance in the current oncotherapy landscape titled Multi-Omics and Network Pharmacology Identify IGFBP1 as an m6A-Epigenetic Target of Pueraria in NSCLC Therapy.

Overall, the strategic approach was novel in the manuscript. I am supporting the publication of this article in PLOS Computational Biology. However, before this is possible, this article will need some serious revisions.

The authors reported the Identification of targets for cancer. The dysregulation of N6-methyladenosine (m6A) modification drives progression in non-therapeutics

Few of the cases author needs to modify the author grammatical errors.

Based on the sequence anayiss, MD Simulation, Docking reviewd suggest that some merit in the paper.

This is an interesting research manuscript by Rui Li and co-workers on a topic of growing importance in the current oncotherapy landscape titled Multi-Omics and Network Pharmacology Identify IGFBP1 as an m6A-Epigenetic Target of Pueraria in NSCLC Therapy.

Originality: Excellent

Technical Quality: Good

Clarity of Presentation: Fair

Rate the overall importance of this paper to the field of Medicinal Chemistry: Top 10%

Recommended after minor revision

**Have the authors made all data and (if applicable) computational code underlying the findings in their manuscript fully available?**

Reviewer #1: Yes

Reviewer #2: Yes

PLOS authors have the option to publish the peer review history of their article (what does this mean? ). If published, this will include your full peer review and any attached files.

**Do you want your identity to be public for this peer review?** For information about this choice, including consent withdrawal, please see our Privacy Policy .

Reviewer #1: **Yes:** ALI MOGHADAM

Reviewer #2: No

**Figure resubmission:**

**Reproducibility:**



---

## [Decision Letter · Decision Letter 1]

8 Feb 2026

PCOMPBIOL-D-25-01915R1

Multi-Omics and Network Pharmacology Identify IGFBP1 as an m 6 A-Epigenetic Target of Pueraria in NSCLC Therapy

PLOS Computational Biology

Dear Dr. Qu,

Thank you for submitting your manuscript to PLOS Computational Biology. After careful consideration, we feel that it has merit but does not fully meet PLOS Computational Biology's publication criteria as it currently stands. Therefore, we invite you to submit a revised version of the manuscript that addresses the points raised during the review process.

We look forward to receiving your revised manuscript.

Kind regards,

Ferhat Ay, Ph.D

Section Editor

PLOS Computational Biology

Ferhat Ay

Section Editor

PLOS Computational Biology

**Journal Requirements:**

**Reviewers' comments:**

Reviewer's Responses to Questions

**Comments to the Authors:**

Reviewer #2: I have satisfied with the revision comments submitted by the Author. Even though need some minor revisions in the manuscript. I have noticed that many of the recent literatures not been cited by the authors, so I recommend to add few more important refences related to the submitted work to strengthen the quality of the manuscript.

Still manuscript needs to proof read by the English readers. So I am recommending for the minor revision.

**Have the authors made all data and (if applicable) computational code underlying the findings in their manuscript fully available?**

Reviewer #2: Yes

PLOS authors have the option to publish the peer review history of their article (what does this mean? ). If published, this will include your full peer review and any attached files.

**Do you want your identity to be public for this peer review?** For information about this choice, including consent withdrawal, please see our Privacy Policy .

Reviewer #2: **Yes:** SREEKANTH THOTA

**Figure resubmission:**
---

## [Editor Report · Decision Letter 2]

22 Feb 2026

Dear Prof. Qu,

We are pleased to inform you that your manuscript 'Multi-Omics and Network Pharmacology Identify IGFBP1 as an m 6 A-Epigenetic Target of Pueraria in NSCLC Therapy' has been provisionally accepted for publication in PLOS Computational Biology.

Best regards,

Shaun Mahony

Section Editor

PLOS Computational Biology

---

## [Editor Report · Acceptance letter]

PCOMPBIOL-D-25-01915R2

Multi-Omics and Network Pharmacology Identify IGFBP1 as an m 6 A-Epigenetic Target of Pueraria in NSCLC Therapy

Dear Dr Qu,

I am pleased to inform you that your manuscript has been formally accepted for publication in PLOS Computational Biology. Your manuscript is now with our production department and you will be notified of the publication date in due course.

With kind regards,

Anita Estes
